# Rapid whole cell imaging reveals a calcium-APPL1-dynein nexus that regulates cohort trafficking of stimulated EGF receptors

H. M. York [1,2], A. Patil[1,2], U. K. Moorthi[1,2], A. Kaur [3], A. Bhowmik[3], G. J. Hyde [4], H. Gandhi [1,2], A. Fulcher[5], K. Gaus [3,6] & S. Arumugam [1,2,3,6 ✉]

The endosomal system provides rich signal processing capabilities for responses elicited by growth factor receptors and their ligands. At the single cell level, endosomal trafficking becomes a critical component of signal processing, as exemplified by the epidermal growth factor (EGF) receptors. Activated EGFRs are trafficked to the phosphatase-enriched peri-nuclear region (PNR), where they are dephosphorylated and degraded. The details of the mechanisms that govern the movements of stimulated EGFRs towards the PNR, are not completely known. Here, exploiting the advantages of lattice light-sheet microscopy, we show that EGFR activation by EGF triggers a transient calcium increase causing a whole-cell level redistribution of Adaptor Protein, Phosphotyrosine Interacting with PH Domain And Leucine Zipper 1 (APPL1) from pre-existing endosomes within one minute, the rebinding of liberated APPL1 directly to EGFR, and the dynein-dependent translocation of APPL1-EGF-bearing endosomes to the PNR within ten minutes. The cell spanning, fast acting network that we reveal integrates a cascade of events dedicated to the cohort movement of activated EGF receptors. Our findings support the intriguing proposal that certain endosomal pathways have shed some of the stochastic strategies of traditional trafficking and have evolved processes that provide the temporal predictability that typify canonical signaling.

[1] Monash Biomedicine Discovery Institute, Faculty of Medicine, Nursing and Health Sciences, Monash University, Clayton/Melbourne, VIC, Australia. [2] European Molecular Biological Laboratory Australia (EMBL Australia), Monash University, Clayton/Melbourne, VIC, Australia. [3] Single Molecule Science, University of New South Wales, Sydney, Australia. [4] Independent scholar, Sydney, Australia. [5] Monash Micro Imaging, Faculty of Medicine, Nursing and Health Sciences, Monash University, Clayton/Melbourne, VIC, Australia. [6] ARC Centre of Excellence in Advanced Molecular Imaging, UNSW, Sydney, Australia. ✉email: Senthil.arumugam@monash.edu

The endocytic system is critical to a cell's ability to faithfully transduce signals and to process its extracellular environment[1–3]. At the plasma membrane, EGF stimulation results in multiple parallel processes—transient increases in $Ca^{2+}$ [4,5], rapid reorganization of actin filaments[6] and formation of new clathrin-coated pits (CCPs)[7]. EGFR has been contrasted with transferrin in many studies as an example of a ligand-induced, rather than a constitutive receptor, system[8,9]. Stimulated EGFRs initiate new CCPs that are distinct from transferrin receptor-containing CCPs[7,10,11]. Post internalization, the intracellular itineraries of the two receptors are distinct with stimulated EGFRs localizing to late endosomes and transferrin receptors recycled back through the early endosomes marked by Rab5[8,12–16].

Rab5 effectors such as EEA1 and APPL1 and 2 have been shown to be involved in the pre-early endosomal steps of endosomal maturation[17]. APPL1 can bind directly to Rab5[18] as well as to a lipid bilayer via a BAR PH domain[19]. APPL1 has been also shown to bind the cytosolic tail of various receptors, including the adiponectin receptor[20], the nerve growth factor receptor[21] and EGFR[17,22] via its PTB domain. Following endocytosis, EGFR has been shown to enter a population of endosomes marked by APPL1[17,23,24]. It has also been demonstrated that endosomes bearing a subset of clathrin-dependent cargoes, including EGFR, are more mobile and mature faster post internalization[25], compared to those that carry transferrin. To better understand the apparent central role of APPL1 in regulating EGFR, its relation to the observed rapid dynamics of EGF-bearing endosomes, and more generally, the molecular events and processes that lead to intracellular divergence of EGFR and transferrin receptors, we measure the dynamics of the newly formed EGF-bearing endosomes as well as APPL1 within the first 10 min of stimulation by EGF. Capitalizing on the rapid volumetric imaging capabilities of lattice light-sheet microscopy (LLSM)[26], we visualized the first events of internalization of EGF-stimulated receptors and discovered that 1. APPL1 relocates from pre-existing endosomes onto EGFR puncta and 2. APPL1 binding to EGFR results in a cohort, dynein-mediated movement of EGFR towards the peri-nuclear region of the cell where the endosomal maturation and signal quenching of EGFRs occur.

## Results

### APPL1 responds to EGF stimulation by binding immediately to EGFR-bearing endosomes and displays enhanced retrograde motility.

To investigate the whole cell distribution of APPL1 on receptor-bearing endosomes following internalization of fluorescently labeled EGF or transferrin in living HeLa cells transfected with APPL1-EGFP, we took advantage of the more rapid volumetric imaging and decreased photo-bleaching offered by a lattice light-sheet microscope (LLSM)[26]. Fast multi-color, whole-cell imaging revealed the temporal dynamics of all fluorescently labeled endosomal populations. To capture the first cargo-bearing endosomes post-internalization with a time resolution of 4 s for an entire volume of a cell, we devised a setup that allowed us to inject EGF or transferrin as a temporal pulse during image acquisition (Supplementary Fig. 1 and Supplementary Movie 1). 50 μL of 100 nM fluorescently labeled ligand was injected near the imaging region, followed by about 200–300 μL of imaging medium. Thus, in single acquisition sequences of up to 15 min, we could follow the initial binding of ligands to the receptors at the plasma membrane, the appearance of early endosomes formed directly by endocytosis of the ligands, and any of their further movements within the cell; additionally, APPL1 was tracked using APPL1-EGFP. We confirmed that the EGF-bearing endosomes are not macropinosomes and are likely derived from clathrin-mediated endocytosis (Supplementary Fig. 2).

To independently track the coordinates of endosomes labeled with APPL1-EGFP, EGF, or transferrin ligands, we performed 2-color (APPL1-EGFP and Alexa 647-labeled transferrin or EGF) imaging of whole cells at 4 s per volume (Supplementary Movies 1, 2). From the independently tracked coordinates of APPL1-EGFP as compared to those of the two other ligands, we quantified the number of spatio-temporally correlated tracks (see the "Methods" section and Supplementary Fig. 3). Both transferrin and EGF appeared as diffuse background fluorescence immediately following injection, and subsequently appeared as punctate structures on the plasma membrane (Fig. 1a, b). In the case of transferrin, the punctate structures slowly acquired low levels of APPL1, with an average delay of 100 s (Fig. 1b, g and Supplementary Movie 2). In contrast, EGF immediately colocalized with APPL1 at the cell-periphery (Fig. 1a, d), as indicated by the fraction of EGF tracks positive for APPL1 at early time points (gray area, Fig. 1d) and track towards the peri-nuclear region (Supplementary Movie 1).

To further analyze how APPL1-binding affects EGF-bearing endosomes, we developed an analysis workflow that selected the APPL1 tracks that co-tracked with EGF (Fig. 1c). The overall motility characteristics of these tracks were then determined using mean square displacement analysis[6]. In general, upon addition of EGF, the fraction of APPL1-EGF tracks exhibiting directed motion increased slightly while those with constrained motion decreased (Fig. 1e, f). However, by 5 min, of all APPL1 vesicles that showed directed motion, over 80% were also found to be positive for EGF (compare pink/blue curves, Fig. 1e, and columns, Fig. 1e), indicating that APPL1-EGF-bearing endosomes display a strong tendency towards being more mobile (Fig. 1 e, f). In contrast, by 5 min after the addition of transferrin, the APPL1-bearing endosomes did not appear to be perturbed, and showed no distinct alteration in directed motions (Fig. 1h, i). To confirm the role of APPL1 in EGF endosomal trafficking we transfected cells with APPL1 siRNA. This resulted in impaired retrograde trafficking and abrogated peri-nuclear accumulation of EGF assayed by the spatial distribution of EGF positive endosomes compared to control cells (Fig. 1j–l). EGF stimulation also resulted in APPL1-bearing endosomes exhibiting longer tracks, and higher maximum velocities, compared to unstimulated cells (Fig. 2a), as well as displaying prominent retrograde movements en route to the PNR (Supplementary Movie 3).

### EGF-bearing APPL1-positive endosomes utilize dynein to move to the PNR.

To allow tracking at higher resolution we combined LLSM with micropatterning[27]. On micropatterned coverslips with 5 μm lines, cells acquired an elongated shape (see the "Methods" section and Supplementary Fig. 4). We predicted that the elongated shape of the cells would accentuate the directionality of endosomal movements, which, in the case of EGF-bearing APPL1 endosomes, are expected to occur in a PNR-directed, retrograde manner (Fig. 2b). We positioned the lattice light sheet such that an oblique cross-section illuminated the nucleus and adjacent PNR of each cell and acquired images with 200 ms of exposure per frame (Supplementary Movie 4). The number of retrograde movements of APPL1 endosomes in wild-type cells treated with EGF was significantly higher than in unstimulated cells (Fig. 2c, e and Supplementary Movies 5, 6) indicating that EGF stimulation can cause a switch in APPL1 dynamics. While APPL1 is associated with endosomes that exist prior to EGF-stimulation, once it becomes attached to endosomes newly formed by the internalization of EGF, those endosomes are much more likely to move in a retrograde manner. These observations led us to ask three questions: What motors are engaged to enable retrograde movement? What causes APPL1 to switch from pre-existing endosomes to

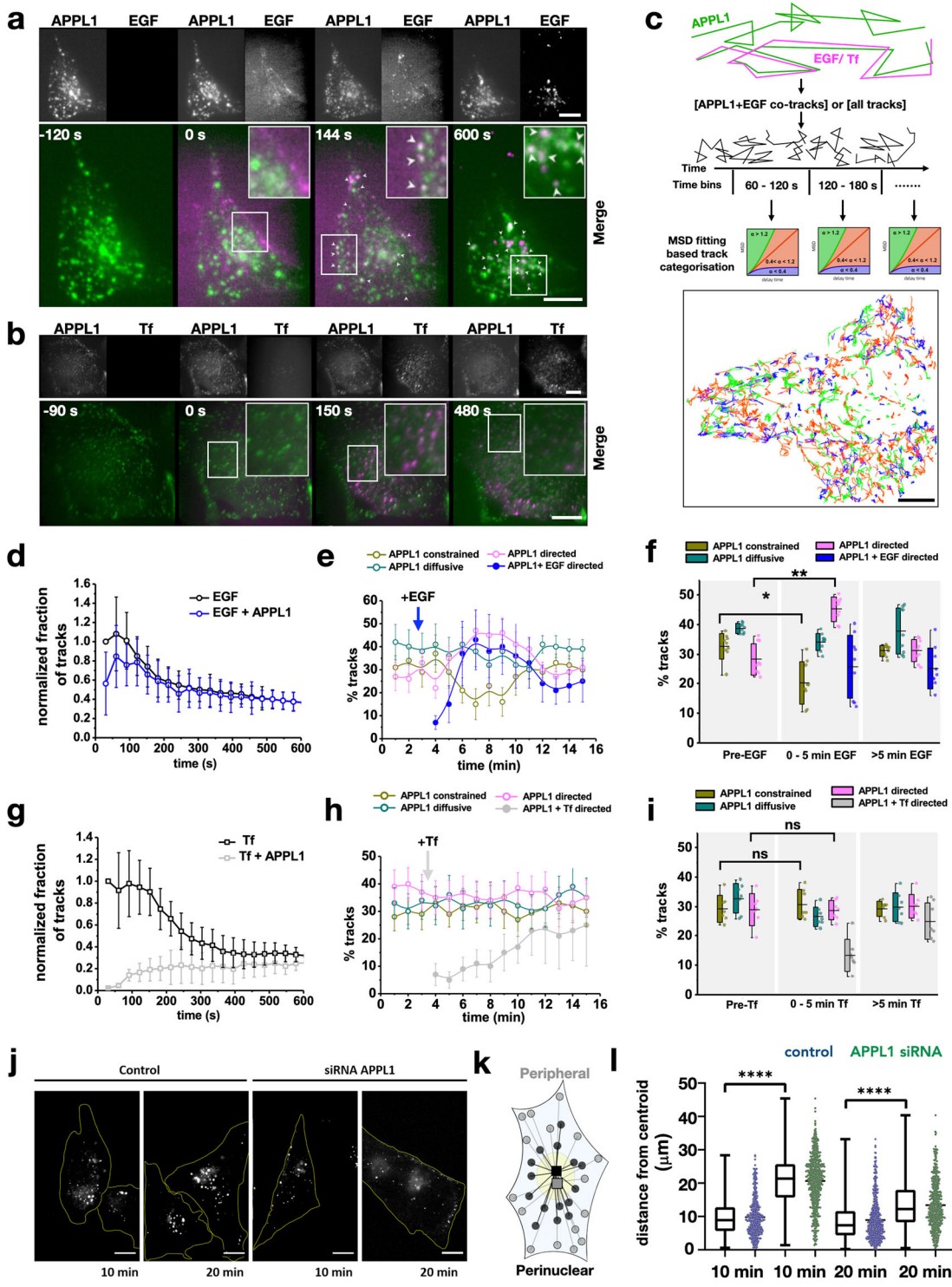

newly generated EGF-bearing endosomes? How does APPL1 localize to EGF-bearing endosomes?

Given the observed retrograde motility of EGF-APPL1-bearing endosomes, we suspected that the minus-end directed motor, dynein, might be involved. Dynein has also been implicated in the translocation of EGF to the juxta-nuclear region in earlier studies[28]. We expressed a fluorescently labeled version of p150 217–548 tagged with DsRed, which sequesters dynein, thus inhibiting dynein-dynactin-based motility[29]. This allowed us to select DsRed-labeled cells, and then assay the motility of their EGF and/or APPL1-bearing endosomes during the first 10 min post

EGF addition (Supplementary Movie 7). Interestingly, while the total set of APPL1-bearing endosomes did not show a significant change in motility, in the subset that co-tracked with an EGF signal, motility was inhibited (Fig. 2f), with a majority of the tracks showing constrained motion (>60%) and only about 15% showing directed motion (Fig. 2f). Inhibiting dynein with ciliobrevin (CB) also resulted in specific inhibition of motility of EGF-APPL1 double positive endosomes, compared to endosomes carrying only APPL1 (Fig. 2d). We quantified the extent of accumulation of endosomes at the PNR by measuring the distance from the centroid, for all endosomal populations (see the "Methods" section

**Fig. 1 EGF rapidly colocalizes with APPL1 and is actively co-trafficked to the perinuclear region. a** Cells transfected with APPL1-EGFP (green) were imaged using LLSM, during which time EGF-A647 (magenta) was pulse injected. The insets correspond to APPL1 and EGFA647 from left to right for the denoted time points. Rectangles indicate zoomed versions for each time point. Time is in seconds. Arrows indicate APPL1 and EGF-A647 colocalization. Scale bar = 10 μm. **b** Similar experiment with fluorescently labeled transferrin. No colocalization was observed. Scale bar = 15 μm. **c** Schematic of tracking and analysis workflow. All identified tracks, or tracks filtered on the basis of co-trafficking by presence of both channels within a determined radius sphere at each time point, were selected for ensemble MSD analysis. Based on the MSD analysis of the tracks, each track was characterized as constrained, diffusive or undergoing directed motion (see the "Methods" section) and the tracks colored correspondingly (bottom) or exported for grouped statistical analysis. **d** Graph of fraction of cargo tracks with time (seconds) following EGF addition to HeLa cells transfected with APPL1 EGFP. Graphs show all the EGF tracks and the fraction of EGF tracks positive for APPL1. Error bars indicate standard deviation ($n = 13$ cells). **e** Percentages of APPL1 tracks categorized as constrained (green), diffusive (teal) and directed (magenta) motions as a function of time by MSD analysis in a single cell. In addition, the blue trace shows the subset of APPL1 tracks positive for EGF that displayed directed motion, demonstrating that most APPL1 tracks with directed motions were also positive for the EGF. Error bars correspond to the standard deviation ($n = 13$ cells). **f** Percentage of APPL1 tracks undergoing constrained (green), diffusive (teal), directed (magenta) motions, and directed motions of EGF-bearing APPL1 endosomes (blue) that were grouped in time as pre-cargo addition, 0–5 and 5–15 min post addition based on example data presented in (**e**). Error bars correspond to the standard deviation where applicable ($n = 9$ cells). **g** Graph of the fraction of cargo tracks with time (s) following transferrin addition to HeLa cells transfected with APPL1 EGFP. Graphs show all the transferrin tracks and the fraction of transferrin tracks positive for APPL1. Error bars indicate standard deviation ($n = 8$ cells). **h** Percentages of APPL1 tracks categorized as constrained (yellow), diffusive (green) and directed (magenta) motions as a function of time by MSD analysis in a single cell. In addition, the graphs show the subset of APPL1 tracks positive for transferrin that display directed motion (gray), demonstrating that only a subset of APPL1 tracks showed the presence of transferrin in the initial time points post-addition. Error bars correspond to the standard deviation ($n = 8$ cells). **i** Percentages of APPL1 tracks undergoing constrained (yellow), diffusive (green), directed (magenta) motions and directed motions of transferrin-bearing APPL1 endosomes (gray) that are grouped in time as pre-cargo addition, 0–5 and 5–15 min post addition based on example data presented in (**e**) Error bars correspond to the standard deviation where applicable ($n = 7$ cells). **j** Representative images of EGF-647 imaged under HILO illumination, 10 and 20-min post-EGF addition in HeLa cells treated with APPL1 siRNA or blank (control). Scale bar = 10 μm. **k** Schematic of endosomal distribution calculation. Euclidean distances between endosomal positions (circles) and the centroid of all the positions (squares) were calculated for cells 10- and 20-min post EGF stimulation. **l** Scatter plots of the endosomal distances from calculated centroids in microns of EGF endosomes in control (blue) and APPL1 siRNA-treated (green) cells as described in **k**. The inner box of the box plot represents the standard deviation, the inner bar the median and the horizontal bars the range. Statistical significance of the difference of the means was evaluated using an unpaired $t$-test, **** represents $p < 0.0001$ ($n = 10$ cells each).

and Fig. 1k). In contrast to controls, no accumulation of EGF vesicles in the PNR was observed when cells expressed p150 217–548 or were treated with dynein inhibitor, CB (Fig. 2h). These results indicate that dynein is the major motor protein involved in the translocation of these endosomes. In agreement with this, the EGF-bearing endosomes exhibited a significantly increased fraction of constrained movements under dynein inhibition, in contrast to the large fraction of directed movements seen in untreated cells (Fig. 2g). Therefore, independent experiments that perturb dynein in distinct ways, both suggest that the accumulation of EGF-APPL1-bearing endosomes in the PNR, via minus-end-directed translocation, requires the recruitment of dynein.

**APPL1-mediated translocation to the perinuclear region is necessary for efficient maturation of EGF-bearing endosomes.** Maturation of APPL1-positive endosomes has been shown to occur through a process of conversion, where one protein is shed off the surface of the endosome, and a new endosomal marker is acquired. APPL1 endosomes convert into EEA1-positive endosomes. This has also been demonstrated for EGFR-containing endosomes[23]. We first recapitulated the maturation of EGF-bearing endosomes (Fig. 3a–d). With time, the number of EGF tracks that had acquired EEA1 increased. Within 5 min, a steady fraction of EGF-EEA1 double positive endosomes is achieved (Fig. 3d). As previously described, conversion processes wherein APPL1 is shed off and EEA1 is acquired were also observed (Fig. 3c)[23]. Inhibiting retrograde motility by treatment of cells with CB slowed down APPL1 to EEA1 maturation of EGF-bearing endosomes (Fig. 3e). The dynein-mediated movement of EGF-APPL1 double positive endosomes towards the PNR is, therefore, necessary for conversion of EGF-APPL1 endosomes into EGF-EEA1 endosomes.

**APPL1 desorption is dependent on a calcium wave evoked by EGF stimulation.** To investigate the observed recruitment of

APPL1, from pre-existing to newly formed EGF-bearing endosomes upon EGF stimulation, we analyzed the whole-cell dynamics of APPL1. Upon stimulation with 100 nM EGF, the APPL1 signal was lost from the pre-existing APPL1-positive endosomes (Fig. 4a, b and Supplementary Movies 8, 9) suggesting global redistribution of APPL1. The pre-existing (pre-EGF-stimulation) APPL1 endosomes were found to be colocalized with Rab5 (Supplementary Fig. 5). We segmented out the PNR to quantify the loss of APPL1 signal from the endosomes. We chose the endosomes at the PNR for this measurement as they are relatively immobile within the time frame of APPL1 desorption, as compared to the peripheral endosomes (Supplementary Movie 10). We found that PNR endosomes lose their APPL1-signal completely or partially when stimulated, respectively, with 100 nM EGF or 20 nM EGF (Fig. 4c and Supplementary Movie 11). In contrast, upon transferrin stimulation, we did not observe any APPL1 redistribution. EGF stimulation has been shown to elicit $Ca^{2+}$ signals through phospholipase cγ (PLCγ) and phospholipase A2 (PLA2)[4,5]. We speculated that the transient increase in cytosolic $Ca^{2+}$ evoked by EGF binding could cause the loss of APPL1 from the endosomes. Previous studies of other proteins that, like APPL1, contain a pleckstrin homology (PH) domain, have shown that elevated intracellular $Ca^{2+}$ prevented membrane binding of such proteins through the formation of Ca–phosphoinositide complexes[30]. To determine if elevated $Ca^{2+}$ is responsible for the unbinding of APPL1, we imaged the dynamic distribution of APPL1 EGFP and the $Ca^{2+}$ indicator R-GECO, after using 100 nM ionomycin to induce a $Ca^{2+}$ influx (Fig. 4d and Supplementary Movie 12)[31]. Upon ionomycin addition, APPL1 dissociated from endosomes within 40 s of the R-GECO signal peak (Fig. 4d). In all cases of ionomycin-mediated desorption, we observed that the endosomes became transiently immobile, after which APPL1 desorbed from them before subsequently rebinding to endosomes. We attribute the freezing of motile endosomes upon an increase in cytosolic $Ca^{2+}$ to 'Calcium-mediated Actin Reset' (CaAR), which results in

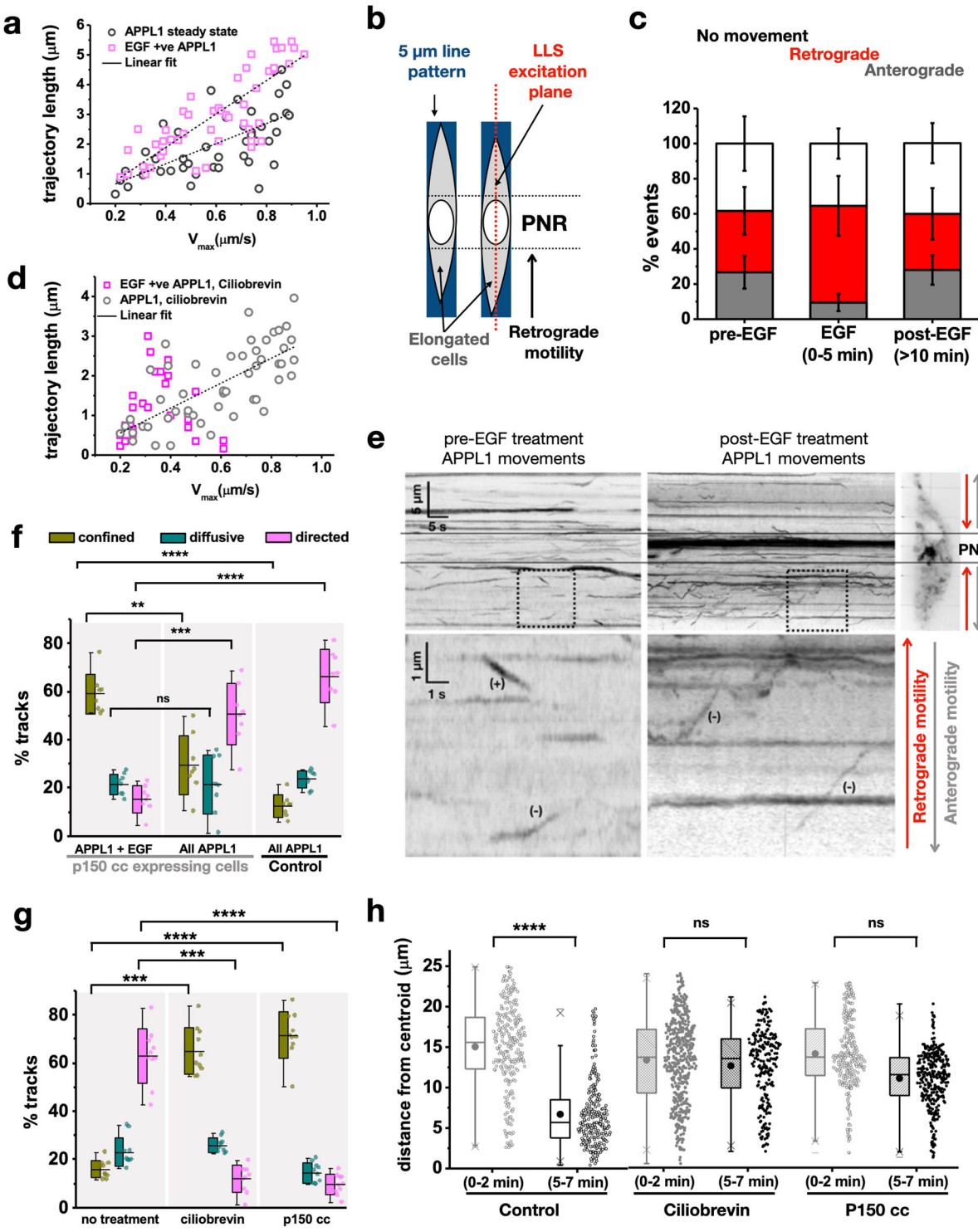

a dense meshwork of actin[32]. We also measured the timing of R-GECO peaks with respect to EGF binding, and APPL1 desorption with respect to EGF binding (Fig. 4e and Supplementary Fig. 6). We found the R-GECO signal peaked about 30 s post EGF stimulation, and APPL1 desorption occurred in the next 30 s (Fig. 4e). The use of calcium-depleted media, with EGTA, did not suppress the intracellular $Ca^{2+}$ rise upon EGF stimulation (Fig. 4f), suggesting ER is the main source of $Ca^{2+}$ upon EGF stimulation, as has been suggested previously[33]. The specific observation of whole cell level APPL1 desorption that occurs within minutes of EGF addition was otherwise not observed when

imaged for long durations or after transferrin addition. Further, the dependency of the extent of APPL1 desorption directly on the concentration of EGF used emphasizes a causal relationship between the two. To verify this, we treated cells with xestospongin-C[34], which blocks store operated calcium channels. We found that for 100 nM EGF treatments, the APPL1 desorption was completely abrogated as demonstrated by lack of intensity loss in trackable endosomes (Fig. 4c) and no relative change in the number of endosomes that can be segmented (Supplementary Fig. 7) as compared to non-treated cells. Together, these results suggest that EGF activation, that results in an

**Fig. 2 Dynein is responsible for rapid EGF-APPL1 trafficking to the perinuclear region. a** Scatter plot of calculated trajectory lengths in microns against maximum velocity in µm/s of APPL1 endosomes in unstimulated cells (black, circles) and EGF-bearing APPL1 endosomes 2–7 min following injection (magenta, squares). Line of best fit was plotted following linear regression. **b** Schematic of single plane lattice illumination (dotted red line) of cells micropatterned in 5 µm patterns (blue). This elongation of the cells accentuates retrograde motility towards the PNR as indicated by the arrow.
**c** Percentages of APPL1 endosomes which underwent anterograde (gray), retrograde (red) or no net movement (white) in micropatterned HeLa cells pre-EGF stimulation, 0–5 min and beyond 10 min post-EGF stimulation. Error bars indicate standard deviation ($n = 13$ cells). **d** Scatter plot of calculated trajectory lengths in microns against maximum velocity in µm/s of APPL1 endosomes in unstimulated cells (gray, circles) and EGF-bearing APPL1 endosomes (magenta, squares) in cells treated with 50 µM ciliobrevin. Line of best fit was plotted following linear regression ($n = 10$ cells). **e** Kymographs of APPL1 endosomes in HeLa cells pre-EGF stimulation (left) and 0–5 min post-EGF stimulation (right). The top graphs show a zoomed-out view, scale bar = 5 µm $y$-axis, 5 s $x$-axis. The overlaid lines show the area of the kymograph corresponding to the peri-nuclear region as seen in the oblique slice inset. The bottom graphs show the area of the dotted box zoomed 5x, '+' indicates plus-end directed motions '−' indicates minus-end directed motions.
**f** Percentages of APPL1-positive endosome tracks which showed confined (green), diffusive (teal), and directed (magenta) motion in HeLa cells 5–15 min post stimulation with 100 nM EGF. The cells were either transfected with DsRed p150 cc or blank DNA (control). Plots correspond to APPL1-EGF double positive endosomes or All APPL1 endosomes. The error bars represent standard deviation ($n = 8$ cells). **g** A bar graph of the percentage of EGF-bearing endosome tracks which show confined (green), diffusive (teal) and directed (magenta) motion in HeLa cells treated with 50 µM ciliobrevin, or transfected with p150 cc DsRed, 5–15 min post stimulation with 100 nM EGF. The error bars represent standard deviation ($n = 10$ cells). **h** Scatter plots of the endosomal distances from centroid in microns (as shown in Fig. 1k), of EGF endosomes in HeLa cells stimulated with 100 nM EGF, at 0–2 and 5–7 min post addition. The dotted box and whisker plots represent cells treated with 50 µM ciliobrevin and the stripped plots cells transfected with DsRed p150 cc ($n = 10$ cells). The inner box of the box plot represents the standard deviation, the inner bar the median and the dot the mean, the 'x' represents the counts within 1–99% of the sample and the horizontal bars the range. Statistical significance of the difference of the means was evaluated using an unpaired $t$-test, **** represents $p < 0.00001$, *** represents $p < 0.0001$, ** represents $p < 0.001$.

increase in intracellular cytosolic calcium, causes the APPL1 to desorb.

**APPL1 binds to EGFR through its PTB domain.** Since it has been previously reported that APPL1 can bind directly to phosphorylated EGFR through its PTB domain[35], we suspected that the conditional localization of APPL1 to EGFR endosomes upon EGF stimulation may be the result of direct binding to the activated EGFR. To verify this, we transfected cells with both EGFP-APPL1 and EGFR-PA TagRFP and visualized the direct interaction between APPL1 and EGFR using FLIM-FRET (fluorescence lifetime microscopy—Försters resonance energy transfer). FLIM-FRET can evaluate direct protein–protein interactions at <6 nm scale while providing spatial resolution at the individual endosome level. Direct interactions result in FRET, which in turn causes a decrease in donor fluorescence lifetime that is very sensitive to direct binding between proteins (Fig. 5a, b). As a control, we used erlotinib-treated cells. We confirmed that erlotinib inhibited EGFR phosphorylation by staining for pEGFR antibody (Supplementary Fig. 8) and that erlotinib treatment abrogated localization of APPL1 to EGF-bearing endosomes (Fig. 5c). Average fluorescence lifetimes of APPL1-eGFP were calculated from regions of interest corresponding to EGFR-PA TagRFP-positive endosomes (>500, from 10 cells). Upon EGF stimulation, the EGFP (donor) tagged to APPL1 showed a significant decrease in fluorescence lifetime (2.12 ± 0.37 ns, mean ± SD) compared to unstimulated conditions, where the EGFP fluorescence lifetime is 2.22 ± 0.24 ns (Fig. 5a, b), similar to values measured in control conditions of cells expressing dominant negative APPL1-ΔPTB-eGFP and EGFR-PA TagRFP treated with EGF (2.28 ± 0.20 ns) or erlotinib-treated cells expressing APPL1-eGFP and EGFR-PA TagRFP (2.27 ± 0.24 ns) (Fig. 5b and Supplementary Fig. 9) which is the native fluorescence lifetime of EGFP measured with cells expressing only APPL1-eGFP (2.23 ± 0.24 ns)[36]. It must be noted that the EGF-treated condition has a much broader distribution with full width at half maxima (FWHM) being significantly broader, at 0.378 ns, compared to that of non-EGF-activated conditions (0.248 ns). We estimate from the values measured in control conditions that this could be due to contributions to the donor lifetimes from both directly interacting APPL1 molecules as well as non-interacting, but endosomal localized, APPL1. To further corroborate that the phosphorylation of EGFR is mediating direct APPL1 binding, cells

expressing WT-APPL1 were treated with 10 µM erlotinib for 1 h, to inhibit EGFR cross-phosphorylation[37], and imaged following EGF stimulation (Fig. 5c). Erlotinib treatment resulted in restricted motility of EGF-bearing endosomes (Fig. 5d) and significantly reduced PNR accumulation (Fig. 5e), with most puncta localized to the periphery of the cell. While the completeness of erlotinib inhibition and its effects on EGFR internalization is debated[38], our experiments show a strong abrogation of association of APPL1 with EGFR, and a reduction in the EGFR localization to PNR in response to erlotinib treatment.

To investigate whether the PTB domain of APPL1 is specifically involved in the binding to EGFR, as suggested elsewhere[22], we expressed a mutant of APPL1 with the PTB domain deleted (APPL1-ΔPTB)[39]. We found that APPL1-ΔPTB did not localize persistently to endosomes in live cell experiments (Supplementary Fig. 9) and the transient punctate interactions may reflect the coincidence detection schemes of BAR proteins[40]. Upon EGF stimulation, APPL1-ΔPTB signals localized to EGF-bearing endosomes transiently with a half-time of 64 s (Supplementary Fig. 9). Furthermore, APPL1-ΔPTB acted as a dominant negative, showing no direct interaction with activated EGFR in FLIM-FRET experiments (Fig. 5b, Supplementary Fig. 10) and strikingly impaired the peri-nuclear localization and directed motion of EGF-bearing endosomes (Fig. 5e). The experiments together suggest that APPL1 is a direct adaptor binding to phosphorylated EGFRs through its PTB domain and mediating dynein-based movement essential for peri-nuclear accumulation of EGF-bearing APPL1-positive endosomes.

## Discussion
Based on the aforementioned experiments that provide evidence of endosomal dynamics and protein redistributions, in Fig. 6 we present a model of the proposed events involving EGF, EGFR, $Ca^{2+}$, APPL1, ER-associated phosphatase, and dynein from before, and up to 5 min after, EGF stimulation. Before exposure to EGF, EGFR predominantly exists in its monomeric state[41], APPL1 is bound to pre-existing endosomes, and $Ca^{2+}$ levels are low. Within 30 s of EGF-EGFR binding, EGFR dimerizes and is phosphorylated, endocytosis generates the first EGF-bearing vesicles, and $Ca^{2+}$ levels begin to rise. Rising $Ca^{2+}$ levels result in the desorption of APPL1 from pre-existing endosomes presumably by blocking the interaction of the BAR and PH domains

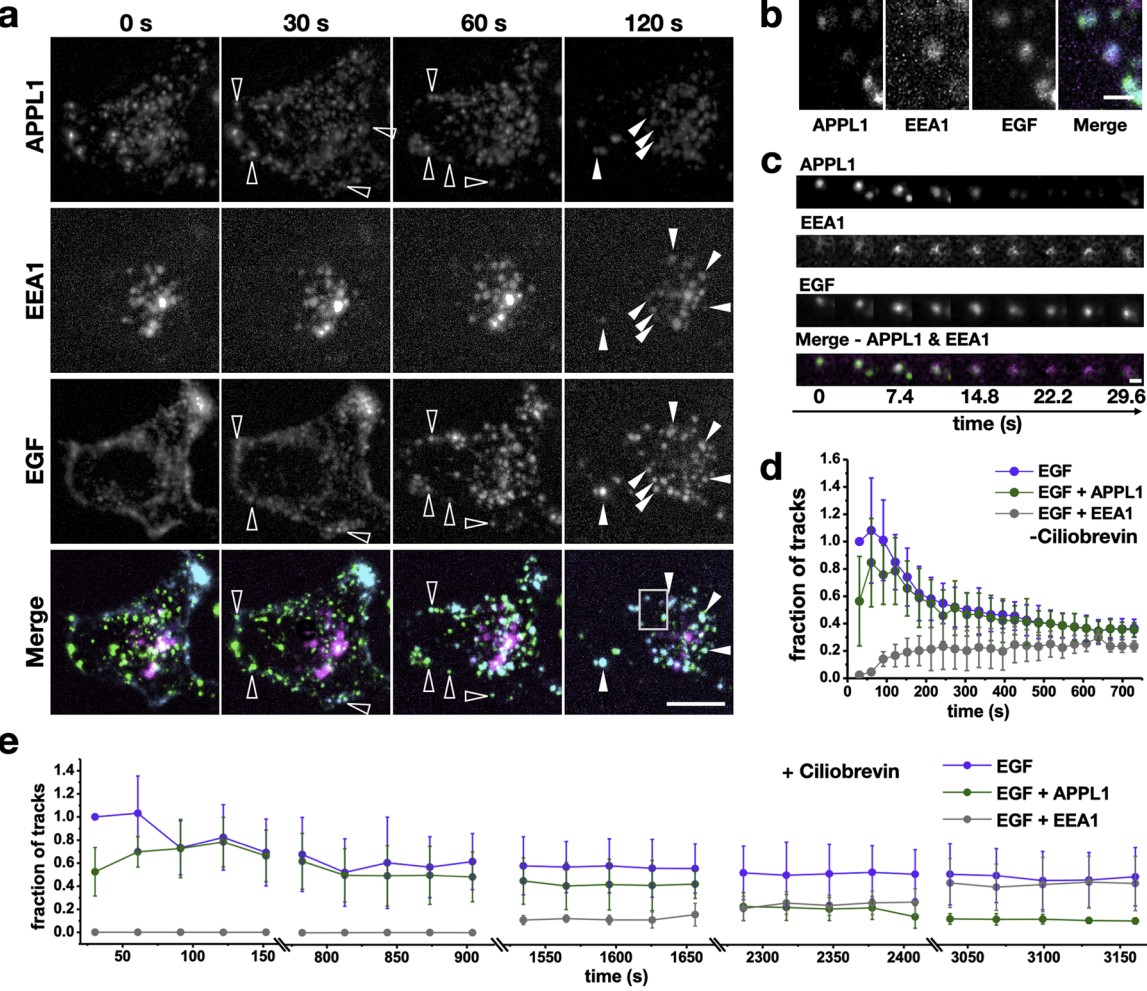

**Fig. 3 APPL1-EGF endosomes mature by acquiring EEA1 and are dependent on its retrograde motility and localization to PNR. a** Representative maximum intensity projections of 100 mM EGF stimulation of HeLa cells expressing APPL1-EGFP and EEA1 TagRFP-T (magenta). Hollow arrows point at a few examples of APPL1 and EGF colocalization and the solid arrows point at APPL1-EEA-EGF1 triple or EEA1-EGF double positive endosomes. Scale bar = 15 µm. **b** A zoomed view of APPL1-EEA1-EGF triple positive endosomes in the boxed region in (**a**). Scale bar = 0.5 µm. **c** An EGF-APPL1 double positive endosome undergoing maturation by conversion to an EGF-EEA1 positive endosome. Scale bar = 0.2 µm. **d** Graph of fraction of EGF tracks positive for APPL1 and EEA1. Error bars indicate standard deviation ($n = 6$ cells). **e** Same plot as (**d**) for cells treated with 50 µM ciliobrevin ($n = 5$ cells). Double lines on the x-axis indicate breaks in time.

of APPL1 and endosomal phosphoinositides, a type of block reported for other PH-domain containing proteins, including Akt[30]. Extracellular EGF concentration controls the degree of $Ca^{2+}$ elevation[5] and in turn, the extent of APPL1 desorption, thereby resulting in a transient availability of cytosolic APPL1 proportional to the EGF concentration as demonstrated by the dependence of APPL1 desorption on EGF concentration. The resulting transient increase in cytosolic APPL1 occurs within ~60 s post EGF-binding, within which time EGFR phosphorylation continues in parallel. Previously, EGFR phosphorylation dynamics, measured using a ratiometric sensor based on EGFR-ECFP and PTB-YFP, revealed that phosphorylation occurred in less than a minute after EGF addition[42]. This is consistent with the dynamics of APPL1 localization to EGFR-bearing endosomes, through the PTB domain, observed in our experiments.

The multiple interaction sites provided by APPL1's BAR, PH, and PTB domains are central to the model. The PTB domain of APPL1 is at the C-terminus and is structurally similar to the PTB domain of Shc[19], which also binds EGFR[43]. The PTB domain of Shc recognizes NPX(phosphor)Y and has been found to bind

tighter to phosphorylated peptides than to phosphatidylinositol[44] and this may explain the preference of APPL1 to bind to phosphorylated EGFR rather than to the membranes through phosphoinositides. Our FLIM-FRET experiments support the idea that the PTB domain of APPL1 binds directly to phosphorylated EGFR. Once APPL1 has bound to EGFR, a form of yet unknown interaction appears to engage dynein, and EGF-bearing endosomes are rapidly shunted to the PNR. Our results suggest that dynein acts downstream to APPL1, in that its activity occurs after APPL1 recruitment, and when APPL1 was blocked from binding to EGFR (by erlotinib or APPL1-ΔPTB), no retrograde motility was seen. Dynein has previously been reported to be required for the translocation of EGF towards the nucleus[28]. While direct interactions between dynein and APPL1 have not been reported, 13 key phosphorylation sites have been identified on APPL1, and their functions are yet to be fully elucidated[45]. How APPL1 coupled to EGFR promotes dynein engagement is unknown, however it is possible that APPL1 phosphorylation sites are involved. APPL1 phosphorylation has previously been implicated in regulating the recycling of activated GPCRs[46]. Collectively

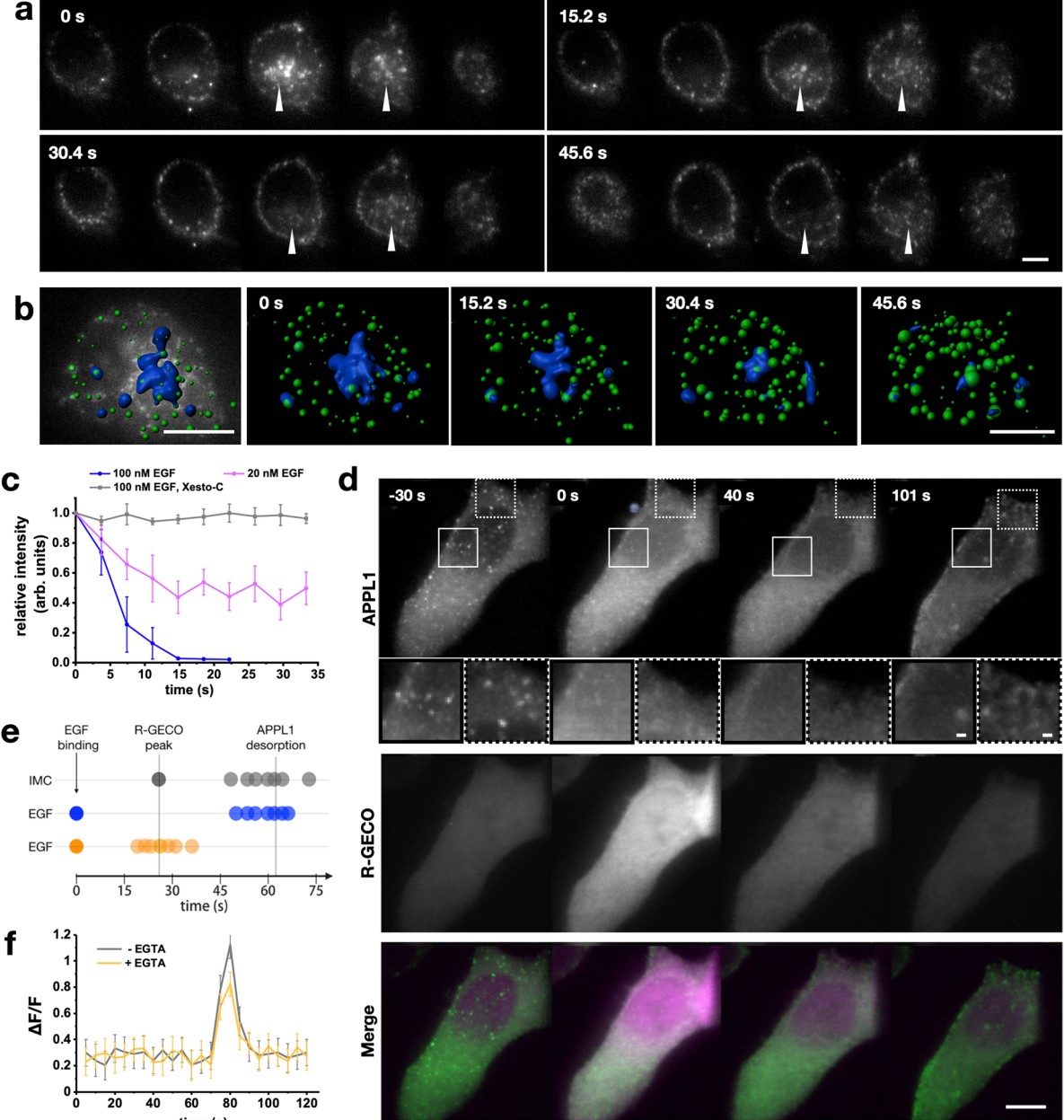

**Fig. 4 EGF-mediated Ca$^{2+}$ wave causes transient APPL1 desorption. a** Time-lapse montage of cross-sections across a single cell displaying desorption of APPL1-EGFP following addition of 100 nM EGF. Scale bar = 4 µm. **b** Example image depicting segmentation of APPL1 EGFP signal in the PNR using Imaris and its loss upon EGF stimulation. Scale bar: 5 µm. **c** Change in relative intensity of APPL1-EGFP in the peri-nuclear region of HeLa cells following addition of an EGF pulse of either 20 nM (green) or 100 nM in untreated (blue) or xestospongin-C treated (gray) cells. Error bars represent standard deviation. ($n = 5$ cells). **d** Maximum intensity projection of images of APPL1-EGFP (top, green in Merge) and R-GECO (middle, magenta in Merge) in HeLa cells stimulated with 100 nM pulse of ionomycin. Time recorded in seconds from ionomycin addition. Scale bar = 10 µm. Insets correspond to zoomed sections (unbroken and dotted squares, respectively) at each time point, scale bar = 1 µm. **e** Timeline of relationships between EGF binding, R-GECO peaks, and APPL1 desorption. Measured R-GECO fluorescence peak and APPL1 desorption (gray dots), each dot represents a single experiment, with R-GECO peaks overlaid to mean of EGF-induced R-GECO peaks. 100 nM EGF was used to determine both the time of APPL1 desorption (blue) and R-GECO peak (orange) ($n = 7$ cells each). **f** R-GECO signal changes upon EGF stimulation in normal media vs. EGTA containing calcium-chelated media ($n = 5$ cells).

these studies highlight the importance of APPL1 phosphorylation and the need for further investigation of the multiple roles of APPL1 in regulating endosomes.

What could be the physiological significance of such rapid trafficking of the EGFR receptors? Endocytosis and trafficking of signaling receptors regulate signal transduction by providing specificity in space and time[3,41]. Endosomal trafficking allows signaling receptors to be localized at specific regions of the cell or to be channeled, in a time-controlled fashion, towards attenuation sites in

multi-vesicular bodies or lysosomes[14]. A recent report by Stanoev et al. demonstrates that EGFR signaling is controlled by protein tyrosine phosphatases (PTPs) associated with ER that are enriched in the PNR[16]. The ER-associated PTPs (PTP1B and TCPTP) enable spatially restricted activity and efficiently de-phosphorylate the EGFR when EGFR localizes to the PNR[12,16]. Our results also agree with the distinct rapid motility previously reported for EGF in contrast to transferrin[25], and reveal the APPL1-mediated engagement of dynein as a mechanism for these movements. This study

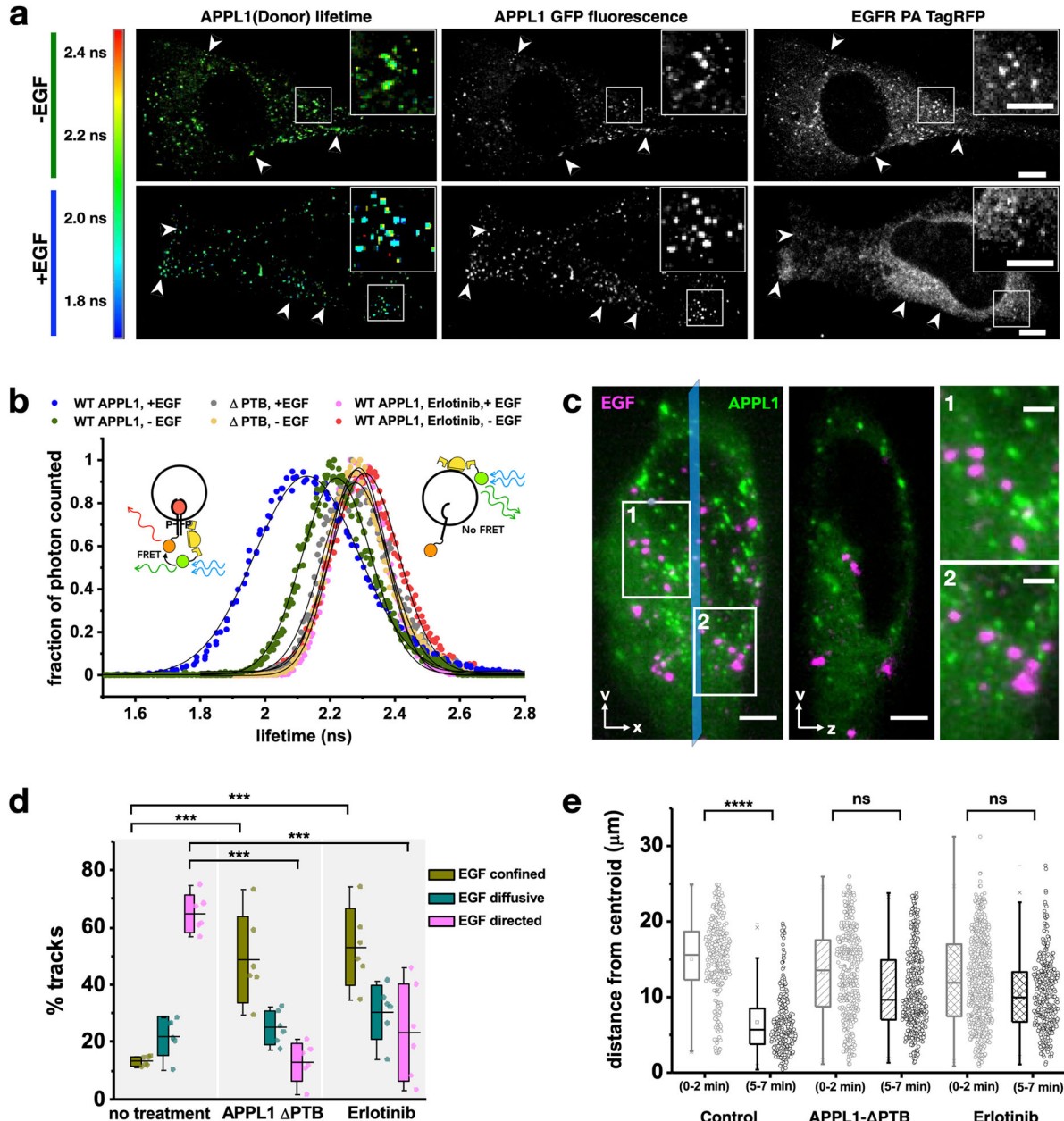

**Fig. 5 APPL1 binds activated EGFR via its PTB domain to mediate minus-directed motility. a** Representative images of FLIM-FRET experiments of HeLa cells transfected with APPL1-EGFP and EGFR-PA-TagRFP, in an unstimulated state and 2 min post 100 nM EGF addition. Colored scale bar represents donor fluorescence lifetime ranging from 1.7 ns (blue) to 2.5 ns (red). Left panels display FLIM image of APPL1 EGFP (donor), center panels display fluorescence image of APPL1 EGFP and right panels display fluorescence image of EGFR-PA-TagRFP. Arrows indicate APPL1-EGFR colocalizing endosomes in all channels, box indicates the region of zoomed inset. Scale bar = 2 μm. **b** Fluorescence lifetimes distribution of APPL1-EGFP photons in unstimulated cells; wild-type APPL1 (green), wild-type APPL1+erlotinib (orange) and APPL1 ΔPTB (yellow). As well as 2 min post 100 nM EGF stimulation in wild-type APPL1 (blue), wild-type APPL1+erlotinib (pink) and APPL1 ΔPTB (gray). Black curves represent a Gaussian fit (n = 8 cells, plot data consists of values from ~400 endosomes for each condition). **c** Representative images of WT-APPL1-GFP (green) in HeLa cells pre-treated with 10 μM erlotinib and stimulated with 100 nM EGF-647 (magenta). Left panel displays a maximum intensity projection (MIP). Center panel displays a cross-sectional slice (YZ view) corresponding to the blue line. No colocalization is observed between EGF and APPL1. Scale bar = 5 μm. Right panels show zoomed regions 1 and 2 from the XY MIP, respectively. Scale bar = 2.5 μm. **d** Percentage of EGF-bearing endosome tracks which showed confined (green), diffusive (teal), or directed (magenta) motility in HeLa cells transfected with APPL1-ΔPTB EGFP or treated with 10 μM erlotinib. Error bars correspond to the standard deviation (n = 6 cells). **e** Scatter plots of EGF distance in microns from centroid in HeLa cells treated with 100 nM EGF at 0–2 and 5–7 min post addition as described in Fig. 1k. The HeLa cells were either transfected with APPL1-ΔPTB EGFP (diagonal lined plot) or treated with 10 μM erlotinib (crossed plot) (n = 10 cells). The inner box of the box plot represents the standard deviation, the inner bar the median and the dot the mean, the 'x' represents the counts within 1–99% of the sample and the horizontal bars the range. Statistical significance of the difference of the means was evaluated using an unpaired t-test, *** represents p < 0.0001, ** represents p < 0.001.

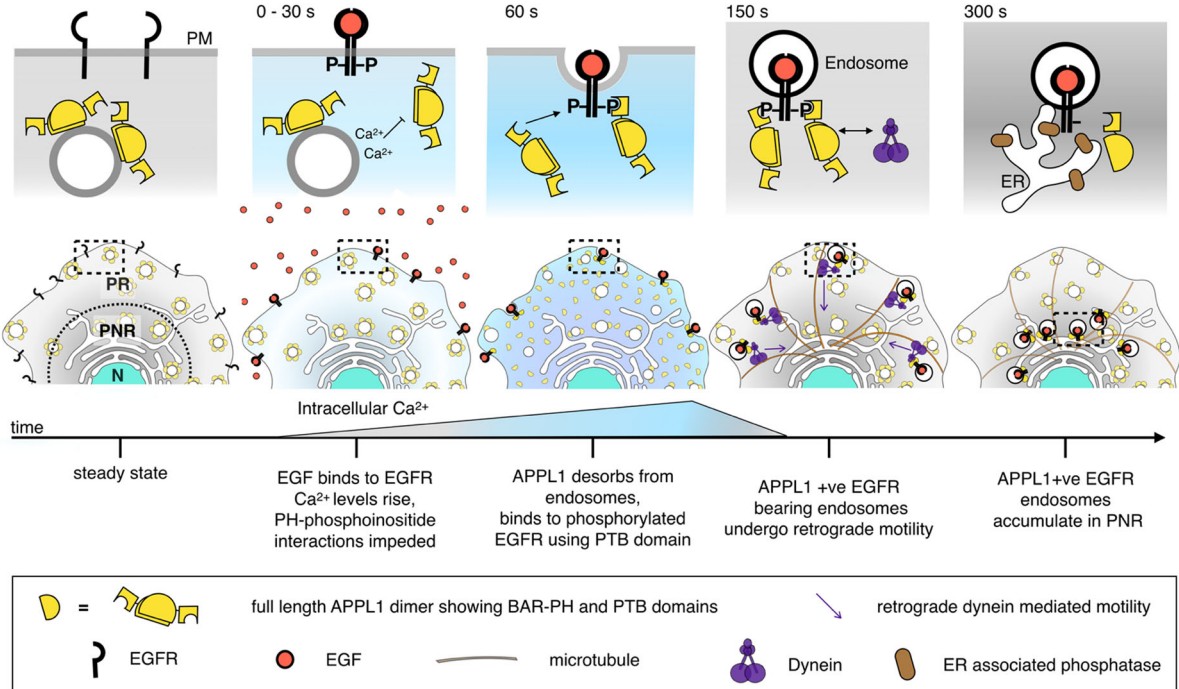

**Fig. 6 Summary figure of proposed mechanism of APPL1-mediated shunt trafficking of activated EGFR.** In steady-state cells, APPL1 (yellow semi-circles) is dispersed throughout the cell. EGF binding leads to dimerization and phosphorylation of EGFR and an increase in intracellular $Ca^{2+}$ (blue background). The increase in intracellular $Ca^{2+}$ impairs APPL1 PH binding to phosphoinositide causing APPL1 desorption. The increased cytosolic APPL1 results in APPL1 binding to activated EGFR via a PTB domain. APPL1 positive EGF-bearing endosomes undergo dynein (purple figure) mediated motility to the peri-nuclear region which is rich in ER.

reveals that APPL1 is central to this dynein-dependent rapid transport, as well as to the peri-nuclear accumulation of stimulated EGFRs and provides a mechanism distinct from that used by constitutive receptors like transferrin. A recent study also reported APPL1 displaying fast retrograde movement in axonal transport hinting that the dynein-based motility that we describe may be crucial in different cell types in different contexts[47].

Vesicular transport is an important logistical support in the signal processing machinery of the cell[48]. As mentioned above, localization of receptors at the PNR, that results in dephosphorylation and hence signal attenuation is an important step in the signal processing. It is likely that not just the localization, but the rate of localization from the periphery of the cell to the PNR will determine the signal sensed. In a scenario where the receptors depend on the steady-state dynamics of the endosomal dynamics, the arrival times of the receptors at the PNR may be broadly distributed. However, as described in this study, if the receptor depends on a specific-activated mechanism where a cohort movement of receptors ensues, the arrival times may be restricted to a narrow distribution, thereby allowing more robust signaling outcomes. This idea is in line with previous studies that have similarly suggested that some endosomal pathways have evolved emergent behaviors to ensure robust signaling[24,49].

Our findings highlight the sophisticated organization of the endosomal system's interaction matrix, where transient interactions driven by specific biochemistry take place in timescales of minutes. Our approach reveals a previously unknown EGFR-APPL1-dynein nexus and highlights how live cell imaging-based studies can unravel transient, conditional interactions, capturing simultaneous multiple processes triggered by a single protein. Such transient interactions are perhaps not discernible in ensemble approaches where temporal resolution, and details of endosomal specificity and dynamics, are lost. Work presented here demonstrates that our whole cell imaging approach can be a

powerful tool in revealing critical transient interactions in key cellular processes such as receptor trafficking.

## Methods

**Cell lines.** HeLa and RPE1 (ATCC) cells were incubated at 37 °C in 5% $CO_2$ in high glucose Dulbecco's modified Eagle's medium (DMEM) (Life Technologies), supplemented with 10% fetal bovine serum (FBS) and 1% penicillin and strepto-mycin (Life Technologies). Cells were seeded at a density of 200,000 per well in a six-well plate containing 5 mm glass coverslips.

**Live-cell imaging.** Cells were imaged using a lattice light sheet microscope (3i, Denver, CO, USA). Excitation was achieved using 488, 560, and 640 nm diode lasers (MPB Communications) at 1–5% AOTF transmittance through an excitation objective (Special Optics 28.6×0.7 NA 3.74-mm immersion lens) and is detected via a Nikon CFI Apo LWD 25×1.1 NA water immersion lens with a ×2.5 tube lens. Live cells were imaged in 8 mL of 37 °C-heated DMEM and were acquired with ×2 Hamamatsu Orca Flash 4.0 V2 sCMOS cameras. 600 µL of 100 nM Alexa647-labeled EGF or transferrin, or 600 µL of 100 µg/µL dextran-fluorescein was added mid-imaging using a custom syringe-sample holder contraption that allowed precise injection of the desired volume of fluorescently labeled ligands, followed by imaging media. The fluorescently labeled ligand containing media were kept separated from the imaging media by an air bubble (Supplementary Fig. 1). Similarly, non-fluorescent 600 µL of 100 nM ionomycin was added where indicated. Cells were serum starved in high glucose DMEM medium 4 h prior to EGF addition.

**Plasmids and transfection.** Cells were transfected with pEGFPC1-human APPL1, a gift from Pietro De Camilli (Addgene plasmid #22198)[23], EGFP-APPL1-ΔPTB, a gift from Donna Webb (Addgene plasmid #59768)[39], DsRed-p150 217-548, a gift from Trina Schroer (Addgene plasmid #51146)[29], EEA1 TagRFP-T, a gift from Silvia Corvera (Addgene plasmid #42635)[50], pEGFR-PATagRFP, a gift from Vladislav Verkhusha (Addgene plasmid #31950)[51]. Cells were transfected with a total of 1 µg DNA (plasmid of interest—0.2 µg, blank DNA—0.8 µg) for single protein expression or (plasmids of interest—0.2 µg + 0.2 µg, blank DNA—0.6 µg) using lipofectamine 3000 (Thermo Fisher Scientific) or the Neon electroporator (Invitrogen) at 1200 mV 20 ms, 2 pulses for RPE1 and 1005 mV 35 ms, 2 pulses for HeLa. HeLa cells were ensured for mild expression by the following: The transfection mix consisted of APPL1 or EEA1 plasmids adding up to 20% of the total DNA amount for transfection, with the rest consisting of blank DNA. Secondly, it has been reported that overexpression of APPL1 or EEA1 results in colocalization of APPL1 and EEA1 on Rab5 endosomes. We optimized this concentration by

screening for this artifact, where we see no overlap of APPL1 and EEA1. Thirdly, APPL1 overexpression impairs EGFR internalization. However, in all our experiments, EGF/EGFR complex is trafficked efficiently to the peri-nuclear region.

**Micropatterning.** Coverslips were sonicated with 70% ethanol for 30 min and plasma cleaned for 5 min. Coverslips were incubated with 1 mg/mL poly(L-lysine)–poly(ethylene-glycol) (PLL–g-PEG) in 1× phosphate buffered saline (PBS) at 4 °C overnight. Coverslips were then placed on a chromium mask with 5 μm line patterns and illuminated for 5 min by an ultraviolet lamp. The coverslips were then incubated with 20 μg/mL fibronectin (Thermo Fisher Scientific) in 1× PBS with 0.02% Tween 20 and 0.04% glycerol for 1 h at room temperature. Following which, cells were plated onto the slips and allowed to take the shape desired over 4 h (Supplementary Fig. 4).

**Drug addition.** Cells were incubated with CB or dimethyl sulfoxide (DMSO) at a final concentration of 50 μM in 8 mL DMEM medium for 5 min before and during imaging. Cells were similarly treated with 10 μM erlotinib or DMSO control for 1 h prior to imaging as indicated. Cytosolic calcium increase was inhibited by 3 μM xestospongin C treatment 30 min before imaging.

**Co-tracking analysis.** Imaris 9.2.1 (Bitplane) was used to detect and track vesicles via the spot detection feature following Gaussian filtering 0.1 μm. The track coordinates were exported and analyzed using custom codes developed in-house. Tracks which followed the same trajectory were filtered by the following routine: The errors in the measurement between the two channels as acquired by the two cameras were obtained by diffusing TetraSpeck beads (T7279, Thermo Fisher Scientific). This estimation of error provided the minimum radius for analyzing the 'co-localization' of spots belonging to two distinct trajectories in two different channels. The spots were considered colocalized if they were within a sphere defined by the minimum radius: $|(p - c)|^2 \leq r^2$ and a time filter of 10 consecutive frames, where $p = x, y, z$ coordinates as a function of time for one channel and $c$ for another (Supplementary Fig. 3). The effective radius of colocalization was set to be 500 nm to account for the sequential imaging and any spatial segregation within a single endosome, based on the measurements from diffusing beads. Next, once all the colocalizing spots were identified, using the trajectory ID of the spots, they were filtered using a minimum time window of three frames (12 s). We discarded any tracks that co-tracked for <12 s as they could be due to transient interactions. Tables detailing the number of co-tracks identified with time were exported, averaged for many samples, and plotted using OriginPro (OriginLab).

**Motility analysis.** The track coordinates were exported from Imaris. MSD analysis was performed as described in previous studies[52,53]. Briefly, the identified trajectories were used to generate MSD plots. The MSD curves were fitted with the following equation: $\mathrm{MSD} = \langle R \rangle^2 = 4Dt^\alpha + 2\sigma^2$, where $D$ is the diffusion coefficient, $t$ is the lag-time, $\alpha$, the scaling exponent, and $\sigma$ is the measurement error measured on the LLSM using beads adsorbed onto glass, fixed as 35 nm for the fits. The trajectories were characterized according to the scaling exponent $\alpha$, retrieved from the fits. Tracks with an exponential value of <0.4 were termed as constrained, those with an exponential value of $0.4 < \alpha < 1.2$ termed diffusive and those with a value >1.2 were said to undergo directed motion.

**Peri-nuclear localization (distance from centroid).** Endosomal distributions within the cell were quantified by calculating the centroid of all endosomal localizations for a given time point. The Euclidean distances of each endosome from this centroid were calculated. These centroid-to-endosome distances were pooled, and ensemble statistical analysis was performed. Peri-nuclear accumulation resulted in lower distances between the centroid and each endosomal localization.

**Fluorescence lifetime imaging.** HeLa cells transfected with EGFP-APPL1 and EGFR-PA TagRFP; fixed with 4% PFA were imaged using a SP8 Falcon (Leica Microsystems) with an 86×1.2NA objective. The samples were photoactivated using 440 nm light using a tunable pulsed white-light laser (10% transmission) for 5 min prior to imaging, PA TagRFP was kept in the fluorescent state using the 440 nm laser at 2% transmission. Fluorescence lifetime images were acquired upon sequential excitation of EGFP-APPL1 and EGFR-PA TagRFP at 488 and 560 nm, respectively, using a tunable pulsed white-light laser (5% transmission) and emission was collected at 495–555 and 565–635 nm using two Leica HyD (Hybrid) detectors. The images were taken such that the brightest pixel reached 2000 photon counts, the lifetimes were then fitted to each pixel using the fast-FLIM mode in the LAS X software (Leica Microsystems).

Samples were also imaged using an FV1000 (Olympus) inverted confocal microscope, with a PicoQuant PicoHarp300 TCSPC system using an Olympus 100×1.4 NA objective. The samples were photoactivated using a 405 nm laser for 5 min prior to imaging, PA TagRFP was kept in the activated state using a 440 nm pulsed laser. Fluorescence images were acquired using 488 and 555 nm excitation on the confocal system. FLIM images were acquired using a 485 nm pulsed laser and a 520/35 nm emission filter, such that each endosomal pixel had over 1000 photon counts. The FLIM images were generated by measuring lifetimes for each

pixel using SymPhoTime 64. Results obtained from both PicoQuant and Leica systems were consistent.

**APPL1 siRNA knockdown.** HeLa cells were transfected with 10 nM APPL1 siRNA with 3 μL Lipofectamine 3000 per well of a six-well plate. The siRNA was sourced from Santa Cruz Biotechnology Inc., and the sequence (5′-CACACCUGACCUCA AAACUTT and 5′-AGUUUUGAGGUCAGGUGUGTT) was based on previously published studies[17]. The cells were then incubated for 48 h prior to Alexa647 labeled EGF addition as described above. Live cells were imaged over the first 10 min following EGF addition as well as cells fixed 10 and 20 min post addition using glutaraldehyde fixation as described by Xu et al.[54]. The cells were imaged using a Nanoimager (Oxford Nanoimaging), under HILO illumination. The cells were excited using a 642 nm laser at 2% transmission and imaged using a ×1.49 NA oil immersion Olympus objective. The endosomes were semi-automatically detected in the imaged planes using the ImageJ plugin TrackMate[55].

**Statistical and reproducibility.** All trajectory data was analyzed blindly. Each experiment contains data from at least $n = 5$ cells and were repeated at least three times. Statistical analysis, performed with OriginPro or GraphPad Prism 8, included all data points obtained for each cell, cells that failed to internalize fluorescent cargo were not included. No test for outliers was employed and the outliers are part of the plots. Data are presented as mean ± standard deviation, unless otherwise noted. Differences in population means was assessed using unpaired $t$-tests, probability ($p$) values < 0.01 were considered as significant (*<0.01, **<0.001, ***<0.0001, ****<0.00001).

**Reporting summary.** Further information on research design is available in the Nature Research Reporting Summary linked to this article.

## Data availability
All raw data are available from the corresponding author on reasonable request.

## Code availability
Custom made codes for endosomal trajectory analysis is available from the corresponding author upon reasonable request or can be found at https://github.com/zeroth/cellphy.

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

## Acknowledgements

This work was supported by the National Health and Medical Research Council of Australia (APP1182212) and ARC LIEF (LE150100163). H.M.Y. is supported by an Australian Government Research Training (RTP) Scholarship. A.P., U.K.M. are supported by Monash Biomedicine Discovery Institute Scholarships. The EMBL Australia Partnership Laboratory (EMBL Australia) is supported by the National Collaborative Research Infrastructure Strategy of the Australian Government. The authors would like to thank Profs. Joe Howard, Marino Zerial for engaging discussions and Drs. Vaishnavi Ananthanarayanan, Angika Basant, and Robert Weatheritt for valuable comments on the manuscript. The authors gratefully acknowledge the Imaging, FACS and Analysis Core and Cameron Nowell at Monash Institute of Pharmaceutical Science for their instrumentation and technical support. The authors acknowledge Monash Micro Imaging, Monash University, for the provision of instrumentation. The authors would like to thank the leadership at Faculty of Nursing, medicine and Health, Monash University, for COVID-safe access and facilitations to the microscope facilities made available to us through 2020 to be able to complete this manuscript.

## Author contributions

S.A. devised the project and directed the study, H.M.Y., U.K.M., A.K., S.A. performed LLSM experiments, H.M.Y., A.F., and S.A. performed fluorescence lifetime experiments. H.M.Y., A.P., S.A. analyzed data. A.B. and U.K.M. performed the micropatterning, H.M.Y., A.K., A.B., U.K.M., H.G. contributed to molecular biology. H.M.Y., U.K.M., H.G., A.P., G.J.H., K.G., and S.A. discussed results, helped shape the research, analysis and manuscript. S.A., H.M.Y., and G.J.H. wrote the manuscript.

## Competing interests

The authors declare no competing interests.
