## [Peer Review File · Communications Biology]

Reviewers' comments:

Reviewer #1 (Remarks to the Author):

The manuscript entitled RAPID WHOLE CELL IMAGING REVEALS AN APPL1-DYNEIN NEXUS THAT REGULATES STIMULATED EGFR TRAFFICKING provides a novel insight in EGFR directed trafficking mediated by APPL1 and Dynein.

The authors focus their observations on 4D state-of-the-art lattice light sheet microscopy, which allows a full cell tracing of endocytic vesicles. The data is conclusive and shows clear evidence of an APPL1-Dynein link directing EGFR-containing endosomes to the perinuclear region. York Ham et al. explore the interconnection dynamics between EGFR activation, calcium peak and APPL recruitment which are of significant relevance in the field; moreover, its association with motor protein Dynein to favor direct sorting to the perinuclear region proposes a mechanism for targeted localization. The use of schemes to picture experimental setups and image processing calculations favor the complete understanding of the results. Though conclusions derived from data are robust and elude over interpretation, there are some experimental issues that need to be addressed:

1. The use of APPL1-GFP transient transfection results in protein overexpression that can lead to protein misslocalization as previously shown for Ras GTPases (Pinilla-Macua, Watkins et al. 2016). Moreover, other authors have shown that APPL1 overexpression decelerates EGF/EGFR transition to perinuclear region (Lee, Hahn et al. 2011). It is essential to determine endogenous APPL1 localization to discard any possible undesired effects resulting from overexpression.
2. Concentration of EGF used for imaging stated in page 2 line 37 does not correlate with the one detailed in methods. This difference is relevant since the final concentration of ligand will determine mechanism for EGFR endocytosis (clathrin dependent or clathrin independent) therefore affecting final fate of the active receptors (Sigismund, Algisi et al. 2013).
3. The authors propose that APPL1-GFP is relocated from pre-existing to newly formed EGFR containing endosomes in response EGF receptor activation. And that this transition is induced by a peak in the cytosolic calcium. However the nature of these pre-existing endosomes is not elucidated. I would be interesting to determine which vesicle compartment is acting as APPL1 reservoir in the absence of active EGFR.
4. Pietro di Camilli and colleagues showed that APPL1 is recruited to pre-early-endosomes immediately after clathrin decoating and that it is lost after the recruitment of EEA1 as the endosome matures (Zoncu, Perera et al. 2009). Please comment on the nature of the endosomes containing EGFR/APPL1-GFP after 15min. Are this endosomes of the same nature as the compartment that serves as APPL1 reservoir prior EGF stimulation? Can the prolonged recruitment of APPL1 be an artifact of overexpression? The use of recycling and early/late endosome markers is encouraged.
5. In figure 1 panels A and B the distribution of APPL1-GFP is significantly different, more perinuclear for panel A (EGF response) and on the cell periphery in panel B (transferrin response). Please comment.
6. The authors should revise figure 2 and figure 4 panel letters and legends so they are in agreement with the main text.
7. Methods section (p.9 line 25) describes the use of PMA, which is not found in the text / figures.

Reviewer #2 (Remarks to the Author):

In the manuscript, York and colleagues employ lattice-sheet microscopy to investigate the trafficking of the EGFR through the APPL1 compartment. They showed that EGF is internalized and rapidly colocalized with APPL1-positive endosomes, at variance with Tf, which is also rapidly internalized, but it is more slowly and less colocalizing with APPL1. The EGF-containing APPL1 endosomes present a directed and persistent motion towards the cell center (dependent on dynein function) as compared to EGF-negative endosomes, suggesting that EGF can cause a shift in APPL1 dynamics. Authors showed that upon EGF, APPL1 redistribute from perinuclear endosomes to the cell periphery, and to EGF-positive endosomes. The increase of intracellular calcium caused by treatment of cells with the cell ionophore Ionomycin has similar impact. As EGF treatment cause an increase of intracellular calcium, therefore the authors link the APPL1 redistribution to the increase of calcium induced by EGF. Finally, they showed that overexpression of an APPL1-deltaPTB mutant impaired the localization of EGF-APPL1 endosomes to the perinuclear region.

The imaging observations herein described are very interesting and carefully performed. Some of the phenotypes are striking and clearly visible in the provided videos (e.g. the redistribution of APPL1 upon EGF or calcium influx by Ionomycin). The model proposed is very appealing. However, it is not fully supported by experimental evidences. The molecular workings of APPL1 in response to EGF is not dissected in depth and most conclusions are based on correlative evidences. Authors should provide experimental prove of the mechanism and the model proposed. Therefore, I recommend publication after solving the issues listed below.

Major Issues:

- 1) Authors should causally link the EGF-induced calcium increase in the cytosol to the APPL1 translocation from the perinuclear region to the periphery, i.e. by following the APPL1 relocation upon EGF in presence or absence of inhibition of calcium release in the cytosol. This could be achieved by inhibiting PLCgamma activity through the use of inhibitors widely used in literature, as for instance U73122, or inhibiting the intracellular and/or extracellular calcium stores. To this aim it would be critical to define what is the source of calcium released in the cytosol upon EGF stimulation, e.g. by measuring with R-GECO the cytosolic calcium level, upon EGF, +/- inhibition of the two major calcium reservoirs in the cell: extracellular (e.g. using EGTA) or intracellular (e.g. using inhibitors of the ER calcium channel, IP3R, as for instance xestospongin C).
- 2) To define the critical role of APPL1 on the motility of EGF-positive endosomes, experiments of functional ablation of APPL1 (e.g. through RNAi) should be performed in parallel to overexpression of APPL1deltaPTB mutant (that might act as a dominant negative by subtracting/relocalizing other critical components).
- 3) To prove their model, authors should demonstrate that in their system APPL1 is able to bind the phosphorylated tail of the EGFR (e.g. by CoIP, pull down or others). There is no data on this in literature, as the two papers cited by the authors (Lee JR et al 2011 and Miaczynska M et al., 2004) do not directly demonstrate the binding of APPL1 to the EGFR tail. This is a critical step in the model and should be demonstrated. Also, to prove the model, it should be shown that APPL1 is able bind to the EGFR upon EGF, that the binding is mediated by the PTB domain and that it is affected by the inhibition of EGF-induced calcium release in the cytosol (e.g. by inhibition of PLCgamma, see point 1).

Other important Issues:

- 1) In Fig.1 E, APPL1-EGF-positive endosomes 'constrained' and 'diffusive' are not shown. Is it because they are not present? Please provide a comment and/or a sentence in the text.
- 2) In Fig.2C please provide a P-value to say that retrograde movement is 'significantly higher' in the EGF 0-5 min) sample
- 3) In Fig. 2F, it would be important to show also data on cells transfected with empty vector control,

in parallel to p150 expressing cells, to directly compare the effects.

4) Fig. 4 is not very convincing.

First, in the case of the APPL1 Δ PTB sample, it would be critical to show the effect of this mutant on the EGF-negative APPL1 endosomes, in parallel to the EGFR-positive once. This represent a critical specificity control of the APPL1 mutant.

Second, the erlotinib data is strange. Indeed, erlotinib treatment by inhibiting the EGFR kinase should also vastly inhibit EGF-induced internalization, and EGF should remain largely at the PM. So, in the Erlotinib-treated cell, there should be no EGF-positive endosomes, or a very limited number of them. Is this the case (It doesn't seem so from the picture provided)? If internalization is inhibited, the erlotinib sample is not very useful. Moreover, it is never showed the effectiveness of erlotinib treatment on EGF-induced EGFR phosphorylation, a control that should be provided.

Minor points:

As already discussed the two papers (Lee JR et al, 2011 and Miaczynska M et al, 2004) do not demonstrate APPL1 binding to the EGFR tail and so text should be corrected (pag. 2 line 16).

2) EGFRs that are internalized upon EGF are not all targeted to the late endosome and lysosome for degradation, but a substantial fraction of them, which varies depending on EGF stimulation and cells context, is also recycled back to the PM through a Rab4- or a Rab11-positive compartment (in this case following the same route of Tf). For a review see Sorkin and Goh, Exp Cell Research, 2009. Please, change the sentence in the introduction accordingly (pag 2, lines 8-10).

3) In Fig. 1H, Tf-loaded APPL1 endosomes lately (after 5 min) acquire directed movements, suggesting a delay, please comment on this issue.

4) In Fig. 2D, Ciliobrevin is used but not described in the text (it is introduced later in Fig. 2h).

Reviewer #3 (Remarks to the Author):

In this manuscript, York and colleagues utilized a lattice light-sheet microscope combined with a pulsed drug delivery system to image and analyze the trafficking of stimulated EGFR in live cells. They revealed an EGFR-APPL1-dynein nexus as an endosomal pathway that is distinct from traditional trafficking. While this work utilizes an elegant, state-of-the-art approach and offers potentially useful insights into receptor trafficking, several key points should be addressed before this study can be considered for publication.

Major points:

(1) To determine if elevated Ca²⁺ is responsible for the unbinding of APPL1, the authors imaged the dynamic distribution of APPL1 and cytosolic Ca²⁺ to uncover temporal correlation of a transient Ca²⁺ influx and the desorption of APPL1 from endosomes. However, this correlation does not support the conclusion "EGF mediated Ca²⁺ influx causes transient APPL1 desorption". To determine the causal relationship between these two events, the authors should perturb EGF mediated Ca²⁺ influx and determine the resulting effects on APPL1 desorption. Furthermore, additional experiments, for example a set of experiments involving APPL1 mutants with defective BAR/PH domains, are needed to support the model that EGF mediated Ca²⁺ influx blocked the interaction between BAR and PH domains of APPL1 and endosomal phosphoinositides, thereby causing transient APPL1 desorption.

(2) Statistical analyses should be performed for all bar graph figures.

(3) The submission is full of typos and mistakes. For example, p3 line17, (Fig. 1a, g) should be (Fig. 1a, d); p4 line26, (Fig. 2d) misquotation or Fig. 2d should be p150 217-548 inhibition data? p5 line11 figure citation should include Fig. 3b; p5 line13 (Fig. 3b) should be (Fig. 3c); p5 line23 (Fig. 3c) should be (Fig. 3d).

Minor points:

(1) In Supplementary Figure 2, the authors showed that HeLa cells do not form macropinosomes to support that HeLa cells is good for EGF internalization study. Please clarify the cell state – are these cells under normal growth condition, starvation condition or with EGF treatment?

(2) Legends for Supplementary Figures: "Summary Figure 2" and "Summary Figure 3" should be "Supplementary Figure 2" and "Supplementary Figure 3".

We would like to thank the reviewers for their thorough reading of our manuscript and detailed criticism of our results and arguments. In the following, we address the specific points raised by each reviewer and hope that our answers are fulfilling. The reviewer comments are written in "italic" and our responses are given in "blue".

Reviewer #1 (Remarks to the Author):

The manuscript entitled RAPID WHOLE CELL IMAGING REVEALS AN APPL1-DYNEIN NEXUS THAT REGULATES STIMULATED EGFR TRAFFICKING provides a novel insight in EGFR directed trafficking mediated by APPL1 and Dynein. The authors focus their observations on 4D state-of-the-art lattice light sheet microscopy, which allows a full cell tracing of endocytic vesicles. The data is conclusive and shows clear evidence of an APPL1-Dynein link directing EGFR-containing endosomes to the perinuclear region. York et al. explore the interconnection dynamics between EGFR activation, calcium peak and APPL recruitment which are of significant relevance in the field; moreover, its association with motor protein Dynein to favor direct sorting to the perinuclear region proposes a mechanism for targeted localization. The use of schemes to picture experimental setups and image processing calculations favor the complete understanding of the results. Though conclusions derived from data are robust and elude over interpretation, there are some experimental issues that need to be addressed:

We thank the reviewer for the positive appraisal, and we hope to address each of the concerns.

1. The use of APPL1-GFP transient transfection results in protein overexpression that can lead to protein misslocalization as previously shown for Ras GTPases (Pinilla-Macua, Watkins et al. 2016). Moreover, other authors have shown that APPL1 overexpression decelerates EGF/EGFR transition to perinuclear region (Lee, Hahn et al. 2011). It is essential to determine endogenous APPL1 localization to discard any possible undesired effects resulting from overexpression.

We thank the reviewer for bringing these studies to our notice. We were extremely careful with our expression levels by using only 20% of total DNA for expressing APPL1. One of the effects of APPL1 overexpression is enhanced colocalization with Rab5 and [[A1, in agreement with Zoncu et al. Cell, 2009. In general, APPL1 and [[A1 only show very little colocalization in the absence of any artefacts. By co-expressing APPL1 and [[A1 supplemented with blank DNA, we found that only 10% of vesicle showed colocalization for APPL1 and [[A1 as compared to over expression with 40 % and larger vesicles. We have now added these results in the supplementary material for clarity. Further, as suggested by the reviewer, Lee et al. describe the effects of APPL1 overexpression that impairs [GF/[GFR internalization and movement of [GF/[GFR to the perinuclear region. However, in our case, with mild overexpression, this behaviour is not observed. In fact, the [GF/[GFR are internalized and are trafficked very robustly to the peri-nuclear region. Given these facts that are contrary to overexpression artefacts, we are confident that our

results are robust and are not overexpression effects. Furthermore, our observations of desorption of APPL1 in the mildly over expressed conditions of APPL1, evident for stimulation by EGF only represents a more challenging condition for the phenomenon as compared to lower non-over expressed levels of APPL1, where desorption would be more easily caused by calcium currents.

2. Concentration of EGF used for imaging stated in page 2 line 37 does not correlate with the one detailed in methods. This difference is relevant since the final concentration of ligand will determine mechanism for EGFR endocytosis (clathrin dependent or clathrin independent) therefore affecting final fate of the active receptors (Sigismund, Algisi et al. 2013).

We apologise for the confusion. Page 2 Line 37 only mentions the ligands that were used, but no concentration. The relevant concentrations are now mentioned in the figure legends, text pertaining to the figures as well as the material and methods. To summarize, we found the phenomena of APPL1 desorption, directed trafficking of EGF/EGFR occur for EGF stimulation at both 20 nM and 100 nM.

3. The authors propose that APPL1-GFP is relocated from pre-existing to newly formed EGFR containing endosomes in response EGF receptor activation. And that this transition is induced by a peak in the cytosolic calcium. However, the nature of these pre-existing endosomes is not elucidated. I would be interesting to determine which vesicle compartment is acting as APPL1 reservoir in the absence of active EGFR.

We performed experiments co-expressing APPL1 with Rab5, EEA1, Rab7, Rab11 to elucidate the nature of APPL1 endosomes before EGF stimulation (Pre-existing endosomes).

APPL1 has been studied extensively in the literature. Our results suggest that while APPL1 does not co-localize with EEA1, but consistent with previously published literature, we see maturation of APPL1 endosomes into EEA1 positive ones (Fig. 3). This lack of overlap between APPL1 and EEA1 also shows that the APPL1 is not overexpressed as APPL1 and EEA1 colocalize under overexpressed conditions (Zoncu *et al.* Cell, 2009). We would like to mention that there has been a plethora of APPL1 studies in the past which have described APPL1 as localized on pre-early endosomes and with Rab5 positive early endosomes in different cells from well-known labs including Prof. Zerial, Prof. Pietro di Camilli, Prof. Donna J Webb and Prof. Marta Międzyńska. These works can be found here: Miaczynska *et al.* Cell, 2004, Goto-Silva *et al.* Scientific reports, 2019, Kim *et al.* Mol. Psychiatry 2016, Zhu *et al.* EMBO J, 2007, Chial *et al.* Traffic, 2008, Diggins *et al.* Biochem Soc Trans., 2017, Diggins *et al.* J Cell Sci., 2018). In strong agreement with these published studies, in our experiments, APPL1 does not colocalize with late endosomes marked by Rab7, but a substantial overlap occurs with Rab5 pre- and post- activation by EGF as expected (Supplementary fig. 5). Our observations of EGF containing APPL1

positive endosomes is also in agreement with previously published studies – Zoncu *et al.* Cell 2009 suggest that they precede EEA1 endosomal populations, and display conversion into EEA1. There are nuances based on how APPL1 binds to EGFR bearing endosomes or directly to EGFR through PTB that are revealed by high resolution FLIM-FRET experiments that we have included in the manuscript and as well as highlighted in our reply below.

4. Pietro di Camilli and colleagues showed that APPL1 is recruited to pre-early-endosomes immediately after clathrin decoating and that it is lost after the recruitment of EEA1 as the endosome matures (Zoncu, Perera et al. 2009). Please comment on the nature of the endosomes containing EGFR/APPL1-GFP after 15min. Are these endosomes of the same nature as the compartment that serves as APPL1 reservoir prior EGF stimulation? Can the prolonged recruitment of APPL1 be an artifact of overexpression? The use of recycling and early/late endosome markers is encouraged.

Pietro di Camilli and colleagues (Zoncu *et al.* Cell 2009), in their experiments, explore different cell lines and various endocytic processes. In figure 1 in their manuscript, they describe endosomal conversion in the context of pinosomes. In figure 2, they show colocalization with Clathrin as well as with EGF containing endosomes. Our tracking studies of clathrin, APPL1 and EGF agree with these results whereby EGF entering through clathrin structures show immediate colocalization with APPL1. However, our results described in this manuscript and unpublished ongoing studies suggest that the nature of binding of APPL1 endosomes are distinct. Whereas in case of macropinosomes, it is mediated through phosphoinositide binding, in case of EGFR, it is through the PTB domain of APPL1 (Fig. 5). The nature of conversions is presumably distinct – in case of PTB domain bound APPL1, it is essentially governed by dephosphorylation of EGFR, that results in unbinding of PTB, concomitant with endosomal conversion by EEA1 acquisition. In the case of Phosphoinositide binding through PH BAR domain of APPL1, it is through the conversion of PI3,4P₂ to PI3P, resulting in FYVE domain of the EEA1 mediating EEA1 binding, as has been described in Zoncu *et al.* Cell, 2009. Additionally, our FRET experiments using lifetime measurements reveal that stimulated EGFR, which is phosphorylated binds to APPL1 directly, compared to non-activated EGFR, where APPL1 binds using PH-BAR domains. This strongly suggests distinct modes of APPL1 binding using PH-BAR or PTB domains.

4. In figure 1 panels A and B the distribution of APPL1-GFP is significantly different, more perinuclear for panel A (EGF response) and on the cell periphery in panel B (transferrin response). Please comment.

This is the result of changes induced by serum starvation in the pre-stimulated state of EGF treated cells. Whilst there is some slight variability in the dynamic localisation of this

compartment, it is important to note that transferrin produces no change in APPL1 localisation in comparison to the dramatic change seen for EGF (Figure 1).

6. The authors should revise figure 2 and figure 4 panel letters and legends so they are in agreement with the main text.

We have checked and corrected the errors concerning to these figures.

7. Methods section (p.9 line 25) describes the use of PMA, which is not found in the text / figures.

We have added the details in the relevant text.

Reviewer #2 (Remarks to the Author):

In the manuscript, York and colleagues employ lattice-sheet microscopy to investigate the trafficking of the EGFR through the APPL1 compartment. They showed that EGF is internalized and rapidly colocalized with APPL1-positive endosomes, at variance with Tf, which is also rapidly internalized, but it is more slowly and less colocalizing with APPL1. The EGF-containing APPL1 endosomes present a directed and persistent motion towards the cell center (dependent on dynein function) as compared to EGF-negative endosomes, suggesting that EGF can cause a shift in APPL1 dynamics. Authors showed that upon EGF, APPL1 redistribute from perinuclear endosomes to the cell periphery, and to EGF-positive endosomes. The increase of intracellular calcium caused by treatment of cells with the cell ionophore Ionomycin has similar impact. As EGF treatment cause an increase of intracellular calcium, therefore the authors link the APPL1 redistribution to the increase of calcium induced by EGF. Finally, they showed that overexpression of an APPL1-deltaPTB mutant impaired the localization of EGF-APPL1 endosomes to the perinuclear region.

The imaging observations herein described are very interesting and carefully performed. Some of the phenotypes are striking and clearly visible in the provided videos (e.g. the redistribution of APPL1 upon EGF or calcium influx by Ionomycin). The model proposed is very appealing. However, it is not fully supported by experimental evidences. The molecular workings of APPL1 in response to EGF is not dissected in depth and most conclusions are based on correlative evidences. Authors should provide experimental prove of the mechanism and the model proposed. Therefore, I recommend publication after solving the issues listed below.

We thank the reviewer for the positive comments, and we hope to have acted constructively upon all the suggestions with our experiments and responses below.

Major Issues:

1) Authors should causally link the EGF-induced calcium increase in the cytosol to the APPL1 translocation from the perinuclear region to the periphery, i.e. by following the APPL1 relocalization upon EGF in presence or absence of inhibition of calcium release in the cytosol. This could be achieved by inhibiting PLC β activity through the use of inhibitors widely used in literature, as for instance U73122, or inhibiting the intracellular and/or extracellular calcium stores. To this aim it would be critical to define what is the source of calcium released in the cytosol upon EGF stimulation, e.g. by measuring with R-GECO the cytosolic calcium level, upon EGF, +/- inhibition of the two major calcium reservoirs in the cell: extracellular (e.g. using EGTA) or intracellular (e.g. using inhibitors of the ER calcium channel, IP3R, as for instance xestospongine C).

We have performed various calcium perturbations as suggested by the reviewers. When we performed experiments using PLC inhibitor, U73122, we discovered that APPL1 localization to endosomes was lost, and the signal appeared cytosolic. We decided to

resort to the other methodologies of calcium perturbation considering U73122 may have off target effects as has been described earlier (Klein *et al.* JBC, 2011), or may in general perturb the phosphoinositide homeostasis, thereby disrupting normal APPL1 localization. Similarly, with Xestospongine-C, which drains ER calcium store resulting in cytosolic Calcium, we saw APPL1 cytosolic as well and not endosomal localized. The general rise in cytosolic calcium may cause this, where Calcium may mask Phosphoinositide-PH BAR interactions. However, with EGTA added to the media we observed that there was a minor decrease in cytosolic R-GECO signal change, in strong agreement with plenty of previously published manuscripts that have very convincingly established that Calcium sources are both internal and external for EGF stimulated cells, but the major contributor would be the ER. We would like to emphasise that the well-studied field of Calcium signalling upon EGF stimulation is not the primary aim of the manuscript.

For reference please see the table below which shows a sample of papers describing the EGF-mediated increase in cytoplasmic Ca²⁺, and the contribution of both intracellular and extracellular sources. We have also included a figure summarising the mechanisms of Ca²⁺ release and influx for clarity.

Paper	Intracellular Stores	Extracellular Stores	Additions
Sawyer and Cohen 1981, Biochemistry	-	Showed requirement of ion channels and extracellular Ca ²⁺	A23187 LaCl ₃
Pandiella et al. 1986, Experimental Cell Research	Were shown to contribute	Were shown to contribute	Ca ²⁺ free, EGTA-containing media
Gilligan et al. 1988 FEBS Letters	Initial response	Sustained Ca ²⁺ response	EGTA
Hughes et al. 1991, Molecular Pharmacology	Produced transient increase in cytosolic Ca ²⁺ Dependent on inositol 1,4,5trisphosphate	Required for sustained increase in cytosolic Ca ²⁺	Thapsigargin Heparin Bradykinin Histamine
Hudson et al. 1994 JBC	Unaffected by sphingosine	Calcium influx was modulated in a dose-dependent manner by sphingolipids	Sphingosine
Fu et al. 1994 Cell Research	Endoplasmic Reticulum, predominant source of Ca ²⁺	Contributed	EGTA, Thapsigargin

Tinhofer et al. 1996 J Biol Chem	Contributed Mediated by a PLC γ and Ras-dependent mechanism	Contributed Mediated by a PLC γ and Ras-dependent mechanism	Thapsigargin Asn ¹⁷ – Ras Mutant Anti-Ras antibody (Y13-259) EGFR- Δ PLC γ 1 _{T,k}
Fleet et al. 1999, Biochemical and Biophysical Research Communications	Unaffected by Annexin IV	Inhibited by Annexin IV	Annexin IV
Li et al. 2003., JBC	Xestospongins C and U73122 inhibit release of calcium	As measured by patch clamp Xestospongins C did not inhibit EGF-induced activation of Ca ²⁺ -conducting channels	Xestospongins C
Tajeddine and Gailly 2012., JBC	Xestospongins B suppressed ER calcium release	Xestospongins B suppressed late Ca ²⁺ entry from external medium	Xestospongins B
Marqueze-Pouey et al. 2014 PLOS One	Stimulated by EGF at both pM and nM conc.	Extracellular influx stimulated only at nM conc.	Charybdotoxin (inhibited sensitive K ⁺ channels that influence pM responses)
Caldieri et al. 2017, Science	Xestospongins C and RTN3 KD suppressed ER calcium release	EGTA suppressed Calcium influx	Xestospongins C EGTA RTN3 KD

Adapted from Pochet et al. 2000, Calcium: The Molecular Basis of Calcium Action in Biology and Medicine.

Additionally, we would like to emphasize that the experiments involving treatment of cells with ionomycin, which floods the intracellular milieu with Calcium, resulting in APPL1 desorption is a positive, conclusive experiment establishing the link between calcium and APPL1 when compared to indirect drug perturbations that seems to normally disrupt APPL1 localization. For e.g. Kang *et al.*, PNAS, 2017, used the same experimental strategy to establish the link between high intracellular Calcium inhibiting membrane localization of PH domains by formation of phosphoinositide-Calcium complexes. Experiments with EGTA secludes Calcium from extracellular milieu as a mechanism and narrow ER as the source of Calcium. Since we have shown that APPL1 only desorbs very specifically within 30 +/- 10s of EGF addition, APPL1 responds to independent intracellular calcium release, and that phosphoinositide – PH Bar interactions can be masked by Calcium ions are known, there is none or little room for any other plausible explanation.

2) To define the critical role of APPL1 on the motility of EGF-positive endosomes, experiments of functional ablation of APPL1 (e.g. through RNAi) should be performed in parallel to overexpression of APPL1deltaPTB mutant (that might act as a dominant negative by subtracting/relocalizing other critical components).

We performed experiments using APPL1 siRNA knockdown to show that in treated cells there is a reduction in the perinuclear directed trafficking of EGF in the first 20 minutes

following EGF addition. These experiments have now been included in the manuscript Figure 1.

3) To prove their model, authors should demonstrate that in their system APPL1 is able to bind the phosphorylated tail of the EGFR (e.g. by CoIP, pull down or others). There is no data on this in literature, as the two papers cited by the authors (Lee JR et al 2011 and Miaczynska M et al., 2004) do not directly demonstrate the binding of APPL1 to the EGFR tail. This is a critical step in the model and should be demonstrated. Also, to prove the model, it should be shown that APPL1 is able to bind to the EGFR upon EGF, that the binding is mediated by the PTB domain and that it is affected by the inhibition of EGF-induced calcium release in the cytosol (e.g. by inhibition of PLCgamma, see point 1).

To show that APPL1 directly binds EGFR using its PTB following EGF-stimulation we utilised Forsters Resonance Energy Transfer (FRET) using Fluorescence Lifetime Imaging, which occurs between spectrally overlapping fluorophores localised within 5 nm of each other. By imaging the FRET-mediated change in fluorescence lifetime we showed that upon EGF stimulation, APPL1 and EGFR are capable of FRET on colocalizing endosomes in contrast to pre-stimulated colocalizing endosomes. This is in agreement with the multiple binding modality model of APPL1 that we put forward; in which in the unstimulated state APPL1 is bound to endosomes via a PH-BAR mediated interaction and following EGF stimulation directly binds to phosphorylated EGFR via its PTB domain. We have now included these experiments in the manuscript. Further, the PTB domain of APPL1 to EGFR has been shown by protein microarray studies as well as pull downs in earlier literature. We have included here the relevant figures from the publications. Therefore, we resorted to FLIM-FRET, which gives us 3-6 nanometres of resolution in protein-protein interactions at single endosome levels rather than the bulk characteristics that the pull-down captures.

As pointed out by the reviewer, the APPL1 PTB binding to EGFR is a critical step in our model. While at two instances (see below) PTB binding has been implicated in EGFR binding by pull down (Zhou *et al. Science Bulletin*, 2016) as well as systems wide interactome studies (Jones *et al. Nature*, 2006), [images below] we clearly show using high-resolution FLIM FRET, the interactions at the level of endosomes as well as the dynamic nature of this conditional interaction.

Zhou et al. Science Bulletin, 2016

Other important Issues:

1) In Fig. 1 E, APPL1-EGF-positive endosomes 'constrained' and 'diffusive' are not shown. Is it because they are not present? Please provide a comment and/or a sentence in the text.

With EGF stimulation, all detectable pools of EGF- APPL1 display motility, and constrained and diffusive form very small pools that only arise at later time points. We have mentioned this detail now in the main text.

2) In Fig. 2C please provide a P-value to say that retrograde movement is 'significantly higher' in the EGF 0-5 min) sample

We have added the quantified p-values to the figure.

3) In Fig. 2F, it would be important to show also data on cells transfected with empty vector control, in parallel to p150 expressing cells, to directly compare the effects.

Ok

4) Fig. 4 is not very convincing.

First, in the case of the APPL1 Δ PTB sample, it would be critical to show the effect of this mutant on the EGF-negative APPL1 endosomes, in parallel to the EGFR-positive ones. This represents a critical specificity control of the APPL1 mutant.

We apologise for the confusion in the graph. In this data, we are showing the trajectory analysis of EGF vesicles. In case of APPL1 Δ PTB, there is no colocalization to EGF bearing endosomes. What we aimed to show is the dominant negative effect of APPL1 Δ PTB expression that has an effect on EGF endosome trafficking when compared to WT APPL1. We have also clarified the confusing bits in the text. In the absence of EGF negative, i.e. before adding EGF, APPL1 Δ PTB localizes to endosomes albeit transiently as quantified from LLSM movies (supplementary fig. 8). The interactions of Bar domain proteins with multiple domains is an ongoing study in the lab, and more nuanced results are in the pipeline in line with coincidence detection based schemes for phosphoinositide interactions.

Second, the erlotinib data is strange. Indeed, erlotinib treatment by inhibiting the EGFR kinase should also vastly inhibit EGF-induced internalization, and EGF should remain largely at the PM. So, in the Erlotinib-treated cell, there should be no EGF-positive endosomes, or a very limited number of them. Is this the case (It doesn't seem so from the picture provided)? If internalization is inhibited, the erlotinib sample is not very useful. Moreover, it is never showed the effectiveness of erlotinib treatment on EGF-induced EGFR phosphorylation, a control that should be provided.

The experiments verifying the effectiveness of Erlotinib on EGF- induced EGFR phosphorylation have been now added to the supplementary material (supplementary fig. 8). In our experiments, we do see EGFR puncta upon EGF stimulation. Most EGF puncta are impaired in motility (Fig. 5 e), as well as the distribution towards perinuclear

accumulation remains unchanged (Fig.5f). As seen in our plots, there is a small fraction of tracks even in Erlotinib treated cells. Having scanned the literature on this, we also find that various other studies also show some internalized EGFRs upon Erlotinib treatment. We would like to refer the referees to Tan *et al.* Cell, 2015 (Figures 7f, and S7C). We do not observe a 100% inhibition of internalization, but significant reduction in motility and PNR accumulation.

Minor points:

As already discussed the two papers (Lee JR et al, 2011 and Miaczynska M et al, 2004) do not demonstrate APPL1 binding to the EGFR tail and so text should be corrected (pag. 2 line 16).

2) EGFRs that are internalized upon EGF are not all targeted to the late endosome and lysosome for degradation, but a substantial fraction of them, which varies depending on EGF stimulation and cells context, is also recycled back to the PM through a Rab4-or a Rab11-positive compartment (in this case following the same route of Tf). For a review see Sorkin and Goh, Exp Cell Research, 2009. Please, change the sentence in the introduction accordingly (pag 2, lines 8-10).

3) In Fig. 1H, Tf-loaded APPL1 endosomes lately (after 5 min) acquire directed movements, suggesting a delay, please comment on this issue.

4) In Fig. 2D, Ciliobrevin is used but not described in the text (it is introduced later in Fig. 2h).

Thank you for drawing our attention to these points, we have amended the relevant sections.

Reviewer #3 (Remarks to the Author):

In this manuscript, York and colleagues utilized a lattice light-sheet microscope combined with a pulsed drug delivery system to image and analyze the trafficking of stimulated EGFR in live cells. They revealed an EGFR-APPL1-dynein nexus as an endosomal pathway that is distinct from traditional trafficking. While this work utilizes an elegant, state-of-the-art approach and offers potentially useful insights into receptor trafficking, several key points should be addressed before this study can be considered for publication.

We thank the reviewer for the positive appraisal, and we hope to have addressed the key point below:

Major points:

(1) To determine if elevated Ca²⁺ is responsible for the unbinding of APPL1, the authors imaged the dynamic distribution of APPL1 and cytosolic Ca²⁺ to uncover temporal correlation of a transient Ca²⁺ influx and the desorption of APPL1 from endosomes. However, this correlation does not support the conclusion “EGF mediated Ca²⁺ influx causes transient APPL1 desorption”. To determine the causal relationship between these two events, the authors should perturb EGF mediated Ca²⁺ influx and determine the resulting effects on APPL1 desorption.

We have now quantified the APPL1 desorption and relocation to the periphery under various calcium perturbations as suggested by the reviewers. When we performed experiments using PLC inhibitor, U73122, we discovered that APPL1 localization to endosomes was lost, and the signal appeared cytosolic. We decided to resort to the other methodologies of calcium perturbation considering U73122 may have off target effects as has been described earlier (Klein *et al.* JBC, 2011), or may in general perturb the phosphoinositide homeostasis, thereby disrupting normal APPL1 localization. Similarly, with Xestospongine-C, which drains ER calcium store resulting in cytosolic Calcium, we saw APPL1 not endosomal localized. The general rise in cytosolic calcium may cause this, where Calcium may mask Phosphoinositide-PH BAR interactions. However, with EGTA added to the media we observed that there was a minor decrease in cytosolic R-GECO signal change, in strong agreement with plenty of previously published manuscripts that have very convincingly established that Calcium sources are both internal and external for EGF stimulated cells, but the major contributor would be the ER. We would like to emphasise that the well-studied field of Calcium signalling upon EGF stimulation is not the primary aim of the manuscript.

Please see below for a list of papers that have linked calcium and EGF previously:

For reference please see the table below which shows a sample of papers describing the EGF-mediated increase in cytoplasmic Ca²⁺, and the contribution of both

intracellular and extracellular sources. We have also included a figure summarising the mechanisms of Ca²⁺ release and influx for clarity.

Paper	Intracellular Stores	Extracellular Stores	Additions
Sawyer and Cohen 1981, Biochemistry	-	Showed requirement of ion channels and extracellular Ca ²⁺	A23187 LaCl ₃
Pandiella et al. 1986, Experimental Cell Research	Were shown to contribute	Were shown to contribute	Ca ²⁺ free, EGTA-containing media
Gilligan et al. 1988 FEBS Letters	Initial response	Sustained Ca ²⁺ response	EGTA
Hughes et al. 1991, Molecular Pharmacology	Produced transient increase in cytosolic Ca ²⁺ Dependent on inositol 1,4,5trisphosphate	Required for sustained increase in cytosolic Ca ²⁺	Thapsigargin Heparin Bradykinin Histamine
Hudson et al. 1994 JBC	Unaffected by sphingosine	Calcium influx was modulated in a dose-dependent manner by sphingolipids	Sphingosine
Fu et al. 1994 Cell Research	Endoplasmic Reticulum, predominant source of Ca ²⁺	Contributed	EGTA, Thapsigargin
Tinhofer et al. 1996 J Biol Chem	Contributed Mediated by a PLC γ and Ras-dependent mechanism	Contributed Mediated by a PLC γ and Ras-dependent mechanism	Thapsigargin Asn ¹⁷ – Ras Mutant Anti-Ras antibody (Y13-259) EGFR- Δ PLC γ 1 _{T,k}
Fleet et al. 1999, Biochemical and Biophysical Research Communications	Unaffected by Annexin IV	Inhibited by Annexin IV	Annexin IV
Li et al. 2003., JBC	Xestospongine C and U73122 inhibit release of calcium	As measured by patch clamp Xestospongine C did not inhibit EGF-induced activation of Ca ²⁺	Xestospongine C

		conducting channels	
Tajeddine and Gailly 2012., JBC	Xestospongine B suppressed ER calcium release	Xestospongine B suppressed late Ca ²⁺ entry from external medium	Xestospongine B
Marqueze-Pouey et al. 2014 PLOS One	Stimulated by EGF at both pM and nM conc.	Extracellular influx stimulated only at nM conc.	Charybdotoxin (inhibited sensitive K ⁺ channels that influence pM responses)
Caldieri et al. 2017, Science	Xestospongine C and RTN3 KD suppressed ER calcium release	EGTA suppressed Calcium influx	Xestospongine C EGTA RTN3 KD

Adapted from Pochet et al. 2000, Calcium: The Molecular Basis of Calcium Action in Biology and Medicine.

Additionally, we would like to emphasize that the experiments involving treatment of cells with ionomycin, which floods the intracellular milieu with Calcium, resulting in APPL1 desorption is a positive, conclusive experiment establishing the link between calcium and APPL1 when compared to indirect drug perturbations that seems to normal disrupt APPL1 localization. For e.g. Kang *et al.*, PNAS, 2017, used the same experimental strategy to establish the link between high intracellular Calcium inhibiting membrane localization of PH domains by formation of phosphoinositide-Calcium complexes. Experiments with EGTA secludes Calcium from extracellular milieu as a

mechanism and narrow ER as the source of Calcium. Since we have shown that APPL1 only desorbs very specifically within 30 +/- 10s of EGF addition, APPL1 responds to independent intracellular calcium release, and that phosphoinositide – PH Bar interactions can be masked by Calcium ions are known, there is none or little room for any other plausible explanation.

Furthermore, additional experiments, for example a set of experiments involving APPL1 mutants with defective BAR/PH domains, are needed to support the model that EGF mediated Ca²⁺ influx blocked the interaction between BAR and PH domains of APPL1 and endosomal phosphoinositides, thereby causing transient APPL1 desorption.

APPL1 mutants defective in BA/PH domains, namely, APPL1 carrying R147A, K153A, and K155A is cytoplasmic in its distribution as observed in our experiments as well as others (Broussard *et al.* MBoC, 2012). Therefore, calcium mediated desorption experiments cannot be performed with these mutants.

GFP-APPL1-AAA

GFP-APPL1-AAA

Broussard *et al.* MBoC, 2012

(2) Statistical analyses should be performed for all bar graph figures.
Done.

(3) The submission is full of typos and mistakes. For example, p3 line17, (Fig. 1a, g) should be (Fig. 1a, d); p4 line26, (Fig. 2d) misquotation or Fig. 2d should be p150 217548 inhibition data? p5 line11 figure citation should include Fig. 3b; p5 line13 (Fig. 3b) should be (Fig. 3c); p5 line23 (Fig. 3c) should be (Fig. 3d).

We apologise to the reviewer for the inconvenience caused and very much appreciate the pointing out of these errors and thank the reviewer.

Minor points:

(1) In Supplementary Figure 2, the authors showed that HeLa cells do not form macropinosomes to support that HeLa cells is good for EGF internalization study. Please clarify the cell state – are these cells under normal growth condition, starvation condition or with EGF treatment?

These cells are under serum starvation conditions. We have also repeated by adding EGF along with dextran and see no macropinosomes formation.

(2) Legends for Supplementary Figures: “Summary Figure 2” and “Summary Figure 3” should be “Supplementary Figure 2” and “Supplementary Figure 3”.

We apologise for these errors, the figure legends have been corrected.

Reviewers' comments:

Reviewer #1 (Remarks to the Author):

The revised manuscript from York and colleagues, entitled Rapid Whole Cell Imaging Reveals a Calcium-APPL1-Dynein Nexus That Regulates Cohort Trafficking of Stimulated EGF Receptors, is significantly improved with the addition of a variety of new experimental evidence supporting their initial observations. Experiments showing calcium -linked relocation of APPL1, effect of APPL1 knockdown and APPL1 mutant overexpression strengthen their initial conclusions regarding the key role of APPL-1 in targeted trafficking of EGF-containing endosomes to the perinuclear region. However, there some issues that need to be addressed.

1. APPL1 perinuclear localization prior EGF addition is shown to serve as APPL1 reservoir, and it relocates to peripheral EGF-containing endosomes (Figure 4a). No information about the nature of these perinuclear pool of APPL1 is provided. Is it in the synthetic pathway?

2. The recruitment of APPL1 to endosomes is lost during the maturation of the vesicle and the gain of EEA-1 within 2 minutes after addition of EGF, as shown in figure 3c. Can the authors speculate how is APPL1 recruited to endosomes after 10min of EGF incubation, as shown in figure 1a? Is it transitioning from PTB to PH-BAR binding in the same endosome? Rab 7 negative colocalization is mentioned in the text (p.6 line 27) in pre-early endosomes, but are these EGF-APPL1 positive endosomes after 10 minutes of EGF incubation? Are these late endosomes?

3. No experiment or comment on PMA besides Methods section (p12 line 11-12). Remove from methods or include figure and text.

4. Figures 1f, 1i, 2f, 2g and 5e color code on the bar graph is misleading. The colors mentioned in the text do not match with the colors shown in the graphs.

5. Figures 1f and 1i show two time intervals 0-5 min and >5 min whereas the text states 0-5 min and 5-15 min which is more accurate.

6. Figure 1 legends 1j, 1k, 1l is not in bold.

7. In Figure 3a, 3b it is highly recommended to include high magnification insets showing EGF-APPL1, EGF-EEA1 and EGF-APPL1-EEA1 colocalization. Results would be more conclusive since the colocalization is difficult to see in the whole cell image.

8. In Figure 3d, 3e same time scale is highly recommended to make both graphs easier to compare. Include Fig 3e in main text (p.6 line 16).

Reviewer #2 (Remarks to the Author):

The revised version of the manuscript has improved and the authors have addressed some of the issues that I have raised.

There are still a couple of important issues that have not been solved.

1) The causal relationship between EGF-mediated calcium influx and the APPL1 desorption has not

been demonstrated. The experiments I suggested during the first round of revision were not aimed at investigating "the calcium signaling upon EGF", which - I know - is well studied. Instead, they were aimed at directly link the calcium increase observed upon EGF to APPL1 relocalization, that, at the moment, are only correlative (as also stated by Reviewer 3).

The authors seem to have obtained strange, unexpected, results (not shown) using the inhibitors I suggested, and they provided "ad hoc" explanation for their strange phenotypes. I agree that drugs can have unspecific effects (the same, however, apply to Ionomycin), but there are other ways to interfere with calcium release from the ER (their main hypothesis): for instance, they can ablate the IP3R or the ryanodine receptors. However, I want to point out that the effect of XestosponginC they reported is very strange, i.e. the increase in intracellular calcium. XestosponginC does not "drain the ER calcium stores", as stated by the authors. Indeed, it is an inhibitor of the IP3R channel on the ER surface and it should inhibit the release of calcium from the ER to the cytosol, causing a decrease in cytosolic calcium, if any. Indeed, XestosponginC was previously shown to inhibit the EGF-dependent calcium release from the ER reducing the increase in cytosolic calcium induced by EGF (Delos Santos RC et al, Mol Biol Cell, 2017; Caldieri G et al, Science 2017). I'm wondering whether the authors are employing a too high dose/time of exposure that might cause unspecific effects (XestosponginC is already effective at 3-10 μ M for 30 min).

2) The second point is the result obtained with Erlotinib treatment. EGF-dependent internalization is kinase-dependent and this is well established in the EGFR field (see for instance reviews from Sorkin A lab). So, either the Erlotinib treatment is not completely inhibiting the EGFR phosphorylation (the control provided by the authors seem to suggest that this might be the case; however, IF is not very quantitative and a total pY on EGFR IP would be more informative), or the Erlotinib treatment is inducing the upregulation of a kinase-independent endocytic pathway. In both cases, the result is difficult to be interpreted. I suggest to remove this data.

Reviewer #3 (Remarks to the Author):

The authors have addressed some of the reviewers' comments but there are still a few remaining points:

1. To determine the causal relationship between these two events, the authors should perturb EGF mediated Ca^{2+} influx and determine the resulting effects on APPL1 desorption. BAPTA-AM can be used to lower cytosolic Ca^{2+} .
2. The FLIM-FRET experiments need to be repeated with controls 1) using the APPL1 Δ PTB mutant; 2) with erlotinib treatment; 3) and ideally in the presence of BAPTA-AM. The lifetime of EGFP should be 2.4-2.6 ns, but the lifetime of EGFP under unstimulated condition was determined to be 2.12 ns. Please provide an explanation for this discrepancy and include more details for the FLIM-FRET experiments in Method, including emission band-pass. For image acquisition, it may be better to choose "reach max photons (always 1000)" instead of "10 frame-averaging" because the latter cannot guarantee enough photons collected from dimly fluorescent cells.
3. There are still typos and errors in the revised submission. For example, p4 line 17, "mean square" is repeated. The revised abstract looks incomplete - what do "these factors" refer to in the first sentence? In Fig. 1e & h, there are some stray graphic symbols.

Reviewers' comments:

Reviewer #1 (Remarks to the Author):

The revised manuscript from York and colleagues, entitled Rapid Whole Cell Imaging Reveals a Calcium-APPL1-Dynein Nexus That Regulates Cohort Trafficking of Stimulated EGF Receptors, is significantly improved with the addition of a variety of new experimental evidence supporting their initial observations. Experiments showing calcium-linked relocation of APPL1, effect of APPL1 knockdown and APPL1 mutant overexpression strengthen their initial conclusions regarding the key role of APPL-1 in targeted trafficking of EGF-containing endosomes to the perinuclear region. However, there are some issues that need to be addressed.

1. APPL1 perinuclear localization prior EGF addition is shown to serve as APPL1 reservoir, and it relocates to peripheral EGF-containing endosomes (Figure 4a). No information about the nature of these perinuclear pool of APPL1 is provided. Is it in the synthetic pathway?

The peri nuclear localised APPL1 is more prominently visible to the naked eye as the APPL1 reservoir because they are relatively more immobile, compared to a second, fast moving APPL1 endosomal population. However, we observe that APPL1 desorbs from almost all populations. This is visible in the supplementary movies as well as in the analysis performed, for e.g. in Fig. 4B where the two major populations have been segmented out. We have already outlined in our previous response and in supplementary fig. 5 that most APPL1 endosomes are also positive for Rab5.

2. The recruitment of APPL1 to endosomes is lost during the maturation of the vesicle and the gain of EEA-1 within 2 minutes after addition of EGF, as shown in figure 3c. Can the authors speculate how is APPL1 recruited to endosomes after 10min of EGF incubation, as shown in figure 1a? Is it transitioning from PTB to PH-BAR binding in the same endosome? Rab 7 negative colocalization is mentioned in the text (p.6 line 27) in pre-early endosomes, but are these EGF-APPL1 positive endosomes after 10 minutes of EGF incubation? Are these late endosomes?

Figure 3c shows the time in seconds that it takes for an APPL1+EGF positive endosome to convert into an APPL1+EEA1 positive endosome (30 seconds for the example endosome). The time shown are time in seconds extracted from the beginning of the process and are not absolute time from the addition of EGF. We apologise if the reviewer finds this confusing, but, Fig. 1a clearly shows APPL1 colocalizing to EGF puncta within 2 – 3 min. This is also evident from the ensemble analysis displayed in Fig. 1d-f.

Rab 7 negative colocalization is mentioned in the text (p.6 line 27) in pre-early endosomes, but are these EGF-APPL1 positive endosomes after 10 minutes of EGF incubation? Are these late endosomes?

We again apologise for the confusion, but lines 25-27 clearly states that APPL1 does not colocalise with Rab7 pre-EGF stimulation. Post-EGF stimulation, APPL1 to EEA1 transition may precede or occur in parallel to Rab5 to Rab7 conversions. These are great open questions in the field, and our lab is working on it, and is beyond the scope of this manuscript.

Other minor comments:

3. No experiment or comment on PMA besides Methods section (p12 line 11-12). Remove from methods or include figure and text.
4. Figures 1f, 1i, 2f, 2g and 5e color code on the bar graph is misleading. The colors mentioned in the text do not match with the colors shown in the graphs.
5. Figures 1f and 1i show two-time intervals 0-5 min and >5 min whereas the text states 0-5 min and 5-15 min which is more accurate.
6. Figure 1 legends 1j, 1k, 1l is not in bold.
7. In Figure 3a, 3b it is highly recommended to include high magnification insets showing EGF-APPL1, EGF-EEA1 and EGF-APPL1-EEA1 colocalization. Results would be more conclusive since the colocalization is difficult to see in the whole cell image.
8. In Figure 3d, 3e same time scale is highly recommended to make both graphs easier to compare. Include Fig 3e in main text (p.6 line 16).

We thank the reviewer for pointing out the minor corrections and suggestions. Changes have been made wherever appropriate.

Reviewer #2 (Remarks to the Author):

The revised version of the manuscript has improved and the authors have addressed some of the issues that I have raised.

There are still a couple of important issues that have not been solved.

1) The causal relationship between EGF-mediated calcium influx and the APPL1 desorption has not been demonstrated. The experiments I suggested during the first round of revision were not aimed at investigating “the calcium signaling upon EGF”, which - I know - is well studied. Instead, they were aimed at directly link the calcium increase observed upon EGF to APPL1 relocalization, that, at the moment, are only correlative (as also stated by Reviewer 3).

The authors seem to have obtained strange, unexpected, results (not shown) using the inhibitors I suggested, and they provided “ad hoc” explanation for their strange phenotypes. I agree that drugs can have unspecific effects (the same, however, apply to Ionomycin), but there are other ways to interfere with calcium release from the ER (their main hypothesis): for instance, they can ablate the IP3R or the ryanodine receptors. However, I want to point out that the effect of XestospongineC they reported is very strange, i.e. the increase in intracellular calcium. XestospongineC does not “drain the ER calcium stores”, as stated by the authors. Indeed, it is an inhibitor of the IP3R channel on the ER surface and it should inhibit the release of calcium from the ER to the cytosol, causing a decrease in cytosolic calcium, if any. Indeed, XestospongineC was previously shown to inhibit the EGF-dependent calcium release from the ER reducing the increase in cytosolic calcium induced by EGF (Delos Santos RC et al, Mol Biol Cell, 2017; Caldieri G et al, Science 2017). I’m wondering whether the authors are employing a too high dose/time of exposure that might cause unspecific effects (XestospongineC is already effective at 3-10 uM for 30 min).

We thank the reviewer for pointing out the critical information for Xestospongine C experiments. We performed the experiments using low dose and shorter incubation time (3 μ M Xestospongine-C treatment 30 mins prior to imaging) and indeed, we were able to perform these experiments. Under these conditions, we observed no APPL1 desorption upon EGF addition, demonstrating that EGFR activation is a pre-requisite for APPL1 desorption and relocalization to the EGFRs at the plasma membrane. We would also like to add that we have imaged APPL1 by itself for prolonged durations (~30 min) at 2 seconds per volume, and we see no desorption. This demonstrates that APPL1 by itself does not undergo ensemble desorption or re-localization. Additionally, we do not see this process for Transferrin receptor binding to transferrin. We can add EGF at any instance within the duration of the experiment, and APPL1 desorbs, which we observe across the many instances of repetition we have performed in the lab. With the reviewers suggested experiment that we have added, and by virtue of time-correlated events being an important ingredient in the sense that EGFR activation as a cause is always prior to the effect, i.e. APPL1 desorption, we can confidently say that APPL1 desorption is causally linked and at the very least EGF binding always precedes an event of APPL1 desorption, and can be abrogated upon inhibiting IP3R channels.

2) The second point is the result obtained with Erlotinib treatment. EGF-dependent internalization is kinase-dependent and this is well established in the EGFR field (see for instance reviews from Sorkin A lab). So, either the Erlotinib treatment is not completely inhibiting the EGFR phosphorylation (the control provided by the authors seem to suggest that this might be the case; however, IF is not very quantitative and a total pY on EGFR IP would be more informative), or the Erlotinib treatment is inducing the upregulation of a kinase-independent endocytic pathway. In both cases, the result is difficult to be interpreted. I suggest to remove this data.

We understand that the kinase dependency for EGFR internalization has previously established discrepancies within the published literature. See Madhus and Stang, JCS, 2009 for a brief review. Upon the request of the reviewer, we have performed and repeated our Erlotinib controls, just to verify the activity of a commercial drug, and we find it works as expected (Supplementary Fig. 7). Our results are also in strong agreements with previously published studies such as Jones et al. 2020 and Tan et al. Cell, 2015 as well as Wang, Q., Villeneuve, G. and Wang, Z. (2005). Control of epidermal growth factor receptor endocytosis by receptor dimerization, rather than receptor kinase activation. *EMBO Rep.* 6, 942-948) that directly addresses the issue raised by the reviewer. Furthermore, even if we assume that Erlotinib treatment does not completely block EGFR phosphorylation and thus some residual EGFR molecules are phosphorylated as the reviewer suggests, the drug significantly alters the levels of phosphorylation, internalisation and has an effect on APPL1 co-localisation, binding, motility and peri-nuclear accumulation. Thus, giving further evidence to the specific interaction between activated EGFR and APPL1, which are the major points relevant to this study. We believe, for reasons of veracity and scientific fairness, we will keep these results in the manuscript. Whether kinase activity is absolutely required for EGFR internalization is not the focus of this study, and we shall leave it to other targeted studies on the subject matter.

Reviewer #3 (Remarks to the Author):

The authors have addressed some of the reviewers' comments but there are still a few remaining points:

1. To determine the causal relationship between these two events, the authors should perturb EGF mediated Ca²⁺ influx and determine the resulting effects on APPL1 desorption. BAPTA-AM can be used to lower cytosolic Ca²⁺.

We have now included Xestospongin C experiments to address this, as well as refer to our responses to reviewer # 2's similar concerns.

We were not able to use BAPTA-AM to lower cytosolic Ca²⁺ as recommended as BAPTA-AM has been shown to impair EGFR phosphorylation (Li *et al.* 2011 JBC, Zhang *et al.* Mol. Cell. Biochem. 2007 and Lee *et al.* Am. J. Physiol. Renal Physiol., 2008), which will make any interpretation complex. Again, we would like to emphasize that under no EGF addition, there is no desorption of APPL1 for an imaging duration of 40. Min. Only upon EGF addition does APPL1 desorb, emphasizing the specific nature of this process. EGFR activation is a pre-requisite cause for the APPL1 desorption as an effect. The cleaner Xestospongin C experiments link these time-correlated events as causal events.

2. The FLIM-FRET experiments need to be repeated with controls 1) using the APPL1deltaPTB mutant; 2) with erlotinib treatment; 3) and ideally in the presence of BAPTA-AM. The lifetime of EGFP should be 2.4-2.6 ns, but the lifetime of EGFP under unstimulated condition was determined to be 2.12 ns. Please provide an explanation for this discrepancy and include more details for the FLIM-FRET experiments in Method, including emission band-pass. For image acquisition, it may be better to choose "reach max photons (always 1000)" instead of "10 frame-averaging" because the latter cannot guarantee enough photons collected from dimly fluorescent cells.

The lifetime of eGFP has indeed been shown to be 2.4-2.6 ns in solution (Padilla-Parra *et al.*, 2009 Biophys J) as pointed out by the reviewer. This only holds true for freely diffusing monomers in a solution with refractive index close to water. However, in cells, the lifetime observed is shorter since fluorescence lifetime is sensitive to many factors including refractive index (Tregidgi *et al.* 2008 and Van Manen, Biophysical J, 2008). Further, aggregation of eGFP has been shown to enable homo-FRET and local changes in refractive indices that shorten the life-time result in shorter lifetimes The general histogram of eGFP(wt) stretches from ~2 to ~2.4ns with the peak value mean of 2.29±0.03 ns (See Ghukasyan and Hsu 2010, J Biomedical Optics). Since it is well established that APPL1 binds to endosomes as a dimer (Zhu *et al.* 2007, EMBO J), and since APPL1 is clearly enriched on the surface of the endosomes, our observed lifetimes are well within the expected range and mean. We have now performed this across multiple controls as requested by the reviewer – with cells expressing only APPL1-eGFP, with Erlotinib treatment, and with APPL1 delta PTB. We find that across all these three conditions, we get consistent eGFP lifetimes for APPL1-eGFP. We have now repeated the FLIM experiments acquiring for longer periods to ensure higher photon counts for each pixel (>2000). We were also able to reproduce the results on a different set up than previously described, where total counts for individual pixels could also be monitored, and with better count rates, we get better distributions which also reproduced our previously reported results. These results have been added to the supplementary material as well as the methodology sections have been updated accordingly with details as requested.

3. There are still typos and errors in the revised submission. For example, p4 line 17, "mean square" is repeated. The revised abstract looks incomplete – what do "these factors" refer to in the first sentence? In Fig. 1e & h, there are some stray graphic symbols.

We thank the reviewer for pointing out these errors and have been fixed.

REVIEWERS' COMMENTS:

Reviewer #1 (Remarks to the Author):

The revised manuscript from York and colleagues has improved significantly since the first submission. The addition of some suggested experiments and evidence reinforces the final conclusions, however there are some issues/concerns that would need some clarification.

1. Transferrin is used to compare the effects of APPL1 binding with those observed in activated EGFR. But the data is not completely clear. The authors state that there is no colocalization between APPL1 and transferrin (Figure legend 1B, p.16 lines 27-28) but in the main text and the calculations show some minimal binding. Besides, the images from the panel depicting transferrin internalization and APPL1 recruitment are dim, and the quality is poor when compared with EGFR/APPL1 panel. Also there seems to be an accumulative binding and directed fraction of tracks with APPL1 and transferrin. Can it be accountable to random coexistence of TfR and APPL1 in same or diffraction limited endosomes? How can the authors justify the directionality?

2. Erlotinib and other EGFR tyrosine kinase inhibitors are extremely efficient in blocking kinase function therefore EGFR internalization (see Sorkina et al. J. Biol Chem. 2002). The effect is generally instantaneous and there is no need for the 1h pre-incubation used by the authors. Was the inhibitor kept in the media during the addition of EGF? I agree with reviewer #2 that IF is not as quantitative as EGFR IP or checking EGFR phosphosites like pY1068 or pY1086. Despite the observation of Erlotinib effect of EGF/APPL1 track directionality (Fig. 5d), the endocytosis levels shown in Fig 5c are equivalent to those in absence of Tyrosine Kinase inhibitor. I strongly recommend providing quantitative evidence that EGFR is not active (phosphorylated) in presence of Erlotinib) and add a comparative control (DMSO) that shows endocytosis levels side by side with erlotinib treated cells.

3. There are still some minor typo issues like Supplementary 8 is named Supplementary 7, Fig1 F and I has some of the error bars cut. Supplementary figures 2 and 3 are named as Summary.

4. Please reconsider adding high magnification insets showing EGF-APPL1, EGF-EEA1 and EGF-APPL1-EEA1 colocalization in Figure 3A and 3B. And adjusting time scale in Figure 3D and E. The data would be more visual and comparable.

Reviewer #2 (Remarks to the Author):

I'm satisfied by authors'reply to my issues

Reviewer #3 (Remarks to the Author):

Reviewers' questions have been fully addressed. There are still some typos and errors. Please proofread the entire manuscript one more time.

1. Transferrin is used to compare the effects of APPL1 binding with those observed in activated EGFR. But the data is not completely clear. The authors state that there is no colocalization between APPL1 and transferrin (Figure legend 1B, p.16 lines 27-28) but in the main text and the calculations show some minimal binding. Besides, the images from the panel depicting transferrin internalization and APPL1 recruitment are dim, and the quality is poor when compared with EGFR/APPL1 panel. Also, there seems to be an accumulative binding and directed fraction of tracks with APPL1 and transferrin. Can it be accountable to random coexistence of TfR and APPL1 in same or diffraction limited endosomes? How can the authors justify the directionality?

Please refer to fig. 1 (d, g). We refer to the very first time points post-internalization where, EGF almost instantaneously colocalizes, vs transferrin tracks that only enter APPL1 compartments gradually. This is consistent with previous observations emphasizing the following 1. Transferrin and EGF have distinct transport characteristics (Lakadamyali *et al.* Cell, 2008) and 2. Transferrin does enter APPL1 compartments (Navaroli *et al.* PNAS, 2012 and Kalaidzidis *et al.* JCB, 2015), albeit, at a slower rate as measured by us and by Kalaidzidis *et al.* JCB, 2015. Since our focus is on EGFR in the current study, we do not explore the transport characteristics of transferrin further. APPL1 typically is involved in transport of many cargoes and by virtue of TfR being associated with that compartment, shows directed motion. We do not claim any specific directionality for transferrin receptors here and is outside the scope of this manuscript.

2. Erlotinib and other EGFR tyrosine kinase inhibitors are extremely efficient in blocking kinase function therefore EGFR internalization (see Sorkina et al. J. Biol Chem. 2002). The effect is generally instantaneous and there is no need for the 1h pre-incubation used by the authors. Was the inhibitor kept in the media during the addition of EGF? I agree with reviewer #2 that IF is not as quantitative as EGFR IP or checking EGFR phosphosites like pY1068 or pY1086. Despite the observation of Erlotinib effect of EGF/APPL1 track directionality (Fig. 5d), the endocytosis levels shown in Fig 5c are equivalent to those in absence of Tyrosine Kinase inhibitor. I strongly recommend providing quantitative evidence that EGFR is not active (phosphorylated) in presence of Erlotinib) and add a comparative control (DMSO) that shows endocytosis levels side by side with erlotinib treated cells.

The media also contained Erlotinib during addition of EGF. Our results are also in strong agreements with previously published studies such as Jones et al. 2020 and Tan et al. Cell, 2015 as well as Wang, Q., Villeneuve, G. and Wang, Z. (2005) for a decreased but not complete abrogation of EGFR internalization under Erlotinib treatment. *EMBO Rep.* **6**, 942-948) that directly addresses this issue regarding activity of Erlotinib. As such, the particular issue of specific effect of Erlotinib is not the focus of our study, and we are confident of Erlotinib activity on account of all the differences we see – 1. Lifetime difference for APPL1 binding. 2. Altered transport characteristics and 3. Abrogation of peri-nuclear accumulation and 4. Decreased antibody staining against pEGFR. The specific sites of inhibition are not the subject of this study and is out of scope. We have now included a statement regarding this issue in the manuscript.

3. There are still some minor typo issues like Supplementary 8 is named Supplementary 7, Fig1 F and I has some of the error bars cut. Supplementary figures 2 and 3 are named as Summary.

We thank the reviewer for pointing this out.

4. Please reconsider adding high magnification insets showing EGF-APPL1, EGF-EEA1 and EGF-APPL1-EEA1 colocalization in Figure 3A and 3B. And adjusting time scale in Figure 3D and E. The data would be more visual and comparable.

We have already included a zoomed version that comprised of 3 endosomes and a single zoomed endosome followed through time. The images provided at publication submission are of substantial high resolution and can be zoomed up digitally on a computer screen.